# $k$-MIXUP REGULARIZATION FOR DEEP LEARNING VIA OPTIMAL TRANSPORT

## ABSTRACT

Mixup is a popular regularization technique for training deep neural networks that can improve generalization and increase adversarial robustness. It perturbs input training data in the direction of other randomly-chosen instances in the training set. To better leverage the structure of the data, we extend mixup to $k$-*mixup* by perturbing $k$-batches of training points in the direction of other $k$-batches using displacement interpolation, i.e. interpolation under the Wasserstein metric. We demonstrate theoretically and in simulations that $k$-mixup preserves cluster and manifold structures, and we extend theory studying the efficacy of standard mixup to the $k$-mixup case. Our empirical results show that training with $k$-mixup further improves generalization and robustness across several network architectures and benchmark datasets of differing modalities.

Standard mixup (Zhang et al., 2018) is a data augmentation approach that trains models on weighted averages of random pairs of training points. Averaging weights are typically drawn from a beta distribution $\beta(\alpha, \alpha)$, with parameter $\alpha$ such that the generated training set is *vicinal*, i.e., it does not stray too far from the original dataset. Perturbations generated by mixup may be in the direction of *any* other datapoint instead of being informed by local distributional structure. As shown in Figure 1, this is a key weakness of mixup that can lead to poor regularization when distributions are clustered or supported on an embedded manifold. With larger $\alpha$, the procedure can result in averaged training points with incorrect labels in other clusters or in locations that stray far from the data manifold.

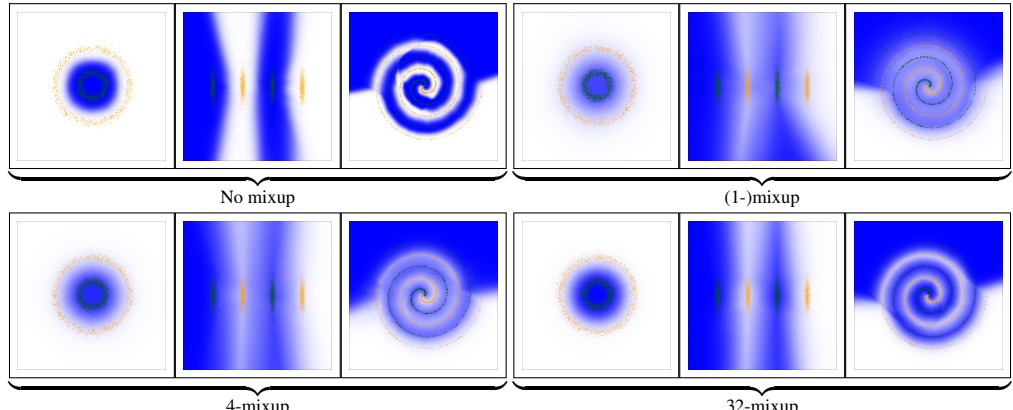

Figure 1: We train a fully-connected network on three synthetic datasets for binary classification, with 1-mixup through 32-mixup regularization ($\alpha = 1$). The plotted functions show the network output and demonstrate that higher $k$-mixup better captures local structure (visible through less blur, increased contrast) while retaining reasonable, even smoothing between the classes.

To address these issues, we present $k$-*mixup*, which averages random pairs of *sets* of $k$ samples from the training dataset. This averaging is done using optimal transport, with *displacement interpolation*. The sets of $k$ samples are viewed as discrete distributions and are averaged as distributions in a geometric sense. If $k = 1$, we recover standard mixup regularization. Figures 1 and 2 illustrate how $k$-mixup produces perturbed training datasets that better match the global cluster or manifold support structure of the original training dataset. Note that the constraints of optimal transport are crucial,   NEW

as for instance a nearest-neighbor approach would avoid the cross-cluster matches necessary for smoothing. In Section 4, we provide empirical results that justify the above intuition.

Our contributions are as follows.

- Empirical results:
    - We show improved generalization results on standard benchmark datasets showing $k$-mixup with $k > 1$ often improves on standard mixup and at worst does no harm.[1]
    - We show that $k$-mixup further improves robustness to adversarial attacks.
- Theoretical characterizations that speak to the claims above in a more precise manner:
    - We argue that as $k$ increases, one is more and more likely to remain within the data manifold (Section 2.1).
    - In the clustered setting, we provide an argument that shows intercluster regularization interpolates nearest points and better smooths interpolation of labels (Section 2.2).
    - We extend the theoretical analysis of Zhang et al. (2020) and Carratino et al. (2020) to our $k$-mixup setting, showing that it leverages local training distribution structure to make more informed regularizations (Section 3).

**Related works.** We tackle issues noted in the papers on adaptive mixup (Guo et al., 2019) and manifold mixup (Verma et al., 2018). The first refers to the problem as "manifold intrusion" and seeks to fix it by training datapoint-specific weights $\alpha$ and considering convex combinations of more than 2 points. This requires additional networks to detect the data manifold and to assign weight combinations, adding substantial complexity to their approach. Manifold mixup deals with the problem by relying on the network to parametrize the data manifold, interpolating in hidden layers of the network. We show in Section 4 that $k$-mixup can be performed in hidden layers to boost performance of manifold mixup. A related approach is that of GAN-mixup (Sohn et al., 2020a)  NEW
which trains a conditional GAN and uses it to generate data points between different data manifolds. As with the other approaches above, the need for training of an additional GAN results in far greater complexity than our $k$-mixup method.

PuzzleMix (Kim et al., 2020) also combines optimal transport ideas with mixup, extending CutMix (Yun et al., 2019) to combine pairs of images. PuzzleMix uses transport to shift saliency regions of images, producing meaningful combinations of input training data. Their use of OT is fundamentally different from ours and does not generalize to non-image data. A recent follow-on work CoMix (Kim et al., 2021), considers a similar approach based on sub/supermodular optimization.

Performing optimal transport between empirical samples of distributions has been considered in  NEW
studies of the *sample complexity* of Wasserstein distance (e.g. Weed & Bach (2019)). Ours is the first application where the underlying source and target distribution are the same, and a theoretical investigation of the generalized notion of $k$-variance is in Solomon et al. (2020). In other works, transport between empirical samples has been dubbed *minibatch optimal transport* and has been used in generative models (Genevay et al., 2018; Fatras et al., 2020) and domain adaptation (Damodaran et al., 2018; Fatras et al., 2021).

## 1 GENERALIZING MIXUP

**Standard mixup.** Mixup uses a training dataset of feature-target pairs $\{(x_i, y_i)\}_{i=1}^N$; the target $y_i$ is a one-hot vector for classification. Weighted averages of training points construct a vicinal dataset:

$$(\tilde{x}_{ij}^\lambda, \tilde{y}_{ij}^\lambda) := (\lambda x_i + (1-\lambda)x_j, \lambda y_i + (1-\lambda)y_j).$$

$\lambda$ is sampled from a beta distribution, $\beta(\alpha, \alpha)$, with parameter $\alpha > 0$ usually small so that the averages are near an endpoint. Using this vicinal dataset, empirical risk minimization (ERM) becomes:

$$\mathcal{E}_1^{mix}(f) := \mathbb{E}_{i,j,\lambda} \left[ \ell \left( f \left( \tilde{x}_{ij}^\lambda \right), \tilde{y}_{ij}^\lambda \right) \right], \quad i,j \sim \mathcal{U}\{1, \ldots, N\}, \lambda \sim \beta(\alpha, \alpha), \tag{1}$$

where $f$ is a proposed feature-target map and $\ell$ is a loss function. Effectively, one trains on datasets formed by averaging random pairs of training points. As the training points are randomly selected,

---

[1]Where $\alpha$ is optimized for both methods.

this construction makes it likely that the vicinal datapoints may not reflect the local structure of the dataset, as in the clustered or manifold-support setting.

$k$-**mixup.** To generalize mixup, we sample two random subsets of $k$ training points $\{(x_i^\gamma, y_i^\gamma)\}_{i=1}^k$ and $\{(x_i^\xi, y_i^\xi)\}_{i=1}^k$. For compactness, let $x^\gamma := \{x_i^\gamma\}_{i=1}^k$ and $y^\gamma := \{y_i^\gamma\}_{i=1}^k$ denote the feature and target sets (and likewise for $\xi$). A weighted average of these subsets is formed with *displacement interpolation* and used as a vicinal training set. This concept is from optimal transport (see, e.g., (Santambrogio, 2015)) and considers $(x^\gamma, y^\gamma)$ and $(x^\xi, y^\xi)$ as uniform discrete distributions $\hat{\mu}^\gamma, \hat{\mu}^\xi$ over their support. In this setting, the optimal transport problem becomes a linear assignment problem (Peyré & Cuturi, 2019). The optimal map is described by a permutation $\sigma \in S_k$ minimizing the cost:

$$W_2^2(\hat{\mu}^\gamma, \hat{\mu}^\xi) = \frac{1}{k} \sum_{i=1}^k \|x_i^\gamma - x_{\sigma(i)}^\xi\|_2^2.$$

Here, $\sigma$ can be found efficiently using the Hungarian algorithm (Bertsimas & Tsitsiklis, 1997). Figure 2 gives intuition for this identification. When compared to the random matching used by standard mixup, our pairing is more likely to match nearby points and to make matchings that better respect local structure—especially by having cross-cluster matches between nearby points on the two clusters.

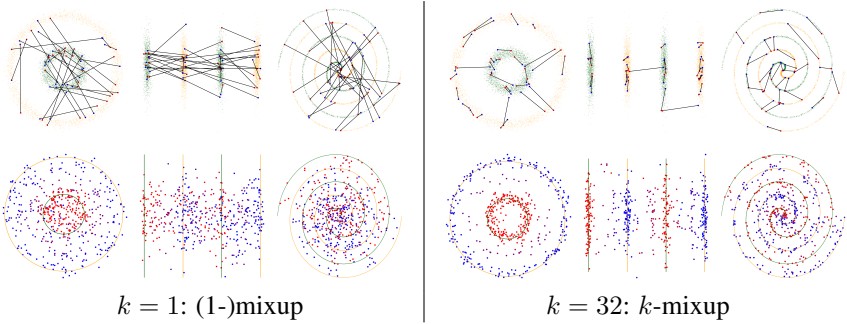

$$k = 1: \text{(1-)mixup} \qquad\qquad k = 32: k\text{-mixup}$$

Figure 2: Optimal transport couplings and vicinal datasets for $k = 1$ (left) and $k = 32$ (right) in 3 simple datasets. In the bottom row, $\alpha = 1$ was used to generate vicinal datasets of size 512.

Given a permutation $\sigma$ and weight $\lambda$, the displacement interpolation between $(x^\gamma, y^\gamma)$ and $(x^\xi, y^\xi)$ is:

$$DI_\lambda((x^\gamma, y^\gamma), (x^\xi, y^\xi)) := \left\{ \lambda(x_i^\gamma, y_i^\gamma) + (1 - \lambda)(x_{\sigma(i)}^\xi, y_{\sigma(i)}^\xi) \right\}_{i=1}^k.$$

As in standard mixup, we draw $\lambda \sim \beta(\alpha, \alpha)$. For the loss function, we consider all $\binom{N}{k}$ subsets of $k$ random samples, i.e., $\{\{(x_i^\gamma, y_i^\gamma)\}_{i=1}^k\}_{\gamma=1}^{\binom{N}{k}}$:

$$\mathcal{E}_k^{mix}(f) = \mathbb{E}_{\gamma, \xi, \lambda} \left[ \ell(f(DI_\lambda(x^\gamma, x^\xi)), DI_\lambda(y^\gamma, y^\xi)) \right], \quad \gamma, \xi \sim \mathcal{U}\{1, \ldots, \binom{N}{k}\}, \lambda \sim \beta(\alpha, \alpha).$$

The localized nature of the matchings makes it more likely that the averaged labels will smoothly interpolate over the decision boundaries. A consequence is that $k$-mixup is robust to higher values of $\alpha$, since it is no longer necessary to keep $\lambda$ close to 0 or 1 to avoid erroneous labels. This can be seen in our empirical results in Section 4, and theoretical analysis appears in Section 3.

**Pseudocode and Computational Complexity** We would like to emphasize the speed and simplicity    NEW
of our method (especially relative to other mixup variants). We have included a brief PyTorch pseudocode in supplement Section A, and note that with CIFAR-10 and $k = 32$ the use of $k$-mixup added 8-9 seconds per epoch. Also in that section is a more extended analysis of computational cost, showing that there is little computational downside to generalizing 1-mixup to $k$-mixup.

## 2 MANIFOLD AND CLUSTER STRUCTURE PRESERVATION

The use of an optimal coupling for producing vicinal data points makes it likely that vicinal datapoints reflect the local structure of the dataset. Below we argue that as $k$ increases, the vicinal couplings will preserve manifold support, preserve cluster structure, and interpolate labels between clusters.

## 2.1 MANIFOLD SUPPORT

Suppose our training data is drawn from a distribution $\mu$ on a $d$-dimensional embedded submanifold $\mathcal{S}$ in $\mathcal{X} \times \mathcal{Y} \subset \mathbb{R}^M$, where $\mathcal{X}$ and $\mathcal{Y}$ denote feature and target spaces. We define an injectivity radius:

**Definition 1** (Injectivity radius). *Let $B_\epsilon(\mathcal{S}) = \{p \in \mathcal{X} \times \mathcal{Y} \mid d(p, \mathcal{S}) < \epsilon\}$ denote the $\epsilon$-neighborhood of $\mathcal{S}$ where $d(p, \mathcal{S})$ is the Euclidean distance from $p$ to $\mathcal{S}$. Define the injectivity radius $R_\mathcal{S}$ of $\mathcal{S}$ to be the infimum of the $\epsilon$'s for which $B_\epsilon(\mathcal{S})$ is not homotopy equivalent to $\mathcal{S}$.*

As $\mathcal{S}$ is embedded, $B_\epsilon(\mathcal{S})$ is homotopy equivalent to $\mathcal{S}$ for small enough $\epsilon$, so $R_\mathcal{S} > 0$. Essentially, Definition 1 grows an $\epsilon$ neighborhood until the boundary intersects itself. We have the following:

**Proposition 1.** *For a training distribution $\mu$ supported on an embedded submanifold $\mathcal{S}$ with injectivity radius $R_\mathcal{S}$, with high probability any constant fraction $1 - \delta$ (for any fixed $\delta \in (0, 1]$) of the couplings induced by $k$-mixup will remain within $B_\epsilon(\mathcal{S})$, for $k$ large enough.*

The proof of this proposition is in supplement Section B. Hence, for large enough $k$, the interpolation induced by optimal transport will approximately preserve the manifold support structure. While the theory requires low dimension $d$ and high $k$ to achieve a tight bound, our empirical evaluations show good performance in the small-$k$ regime. Some schematic examples are shown in Figures 1 and 2.

## 2.2 CLUSTERED DISTRIBUTIONS

With a clustered training distribution we preserve global structure in two ways: by mostly matching points within clusters and by matching (approximately) nearest points across clusters. These characterizations are only true approximately, but are achieved exactly as $k \to \infty$, as argued below.

To refine the definition of a clustered distribution, we adopt the $(m, \Delta)$-clusterable definition used in Weed & Bach (2019); Solomon et al. (2020). In particular, a distribution $\mu$ is $(m, \Delta)$-clusterable if $\text{supp}(\mu)$ lies in the union of $m$ balls of radius at most $\Delta$. Now, if our training samples $(x_i, y_i)$ are sampled from such a distribution, where the clusters are sufficiently separated, then intracluster matchings will be prioritized over cross-cluster matchings under optimal transport.

**Lemma 1.** *Draw two batches of samples $\{p_i\}_{i=1}^N$ and $\{q_i\}_{i=1}^N$ from a $(m, \Delta)$-clusterable distribution, where the distance between any pair of covering balls is at least $2\Delta$. If $r_i$ and $s_i$ denote the number of samples in cluster $i$ in batch 1 and 2, respectively, then the optimal transport matching will have $\frac{1}{2} \sum_i |r_i - s_i|$ cross-cluster matchings.*

The proof of the above (supplement Section C) involves the pigeonhole principle and basic geometry. We also argue that the fraction of cross-cluster identifications approaches zero as $k \to \infty$ and characterize the rate of decrease. The proof (supplement Section D) follows via Jensen's inequality:

**Theorem 1.** *Given the setting of Lemma 1, with probability masses $p_1, \ldots, p_m$, and two batches of size $k$ matched with optimal transport, the expected fraction of cross-cluster identifications is $O\left((2k)^{-1/2} \sum_{i=1}^m \sqrt{p_i(1 - p_i)}\right)$.* NEW

Note that the absolute number of cross-cluster matches still increases as $k$ increases providing more information in the voids between clusters. We emphasize that the $(m, \Delta)$ assumption is designed NEW to create a "worst-case scenario" i.e. a setting where clusters are well-separated such that cross-cluster identifications are as unlikely as possible. Hence in real datasets the fraction of cross-cluster identifications will be larger than indicated in Theorem 1. Finally, we show that these cross-cluster matches are of length close to the distance between the clusters with high probability (proved in supplement Section E), i.e., the endpoints of the match lie in the parts of each cluster closest to the other cluster. This rests upon the fact that we are considering the $W_2$ cost, for which small improvements to long distance matchings yield large cost reductions in squared Euclidean distance.

**Theorem 2.** *Suppose density $p$ has support on disjoint compact sets (clusters) $A$, $B$ whose boundaries are smooth, where $p > 0$ throughout $A$ and $B$. Let $D$ be the Euclidean distance between $A$ and $B$, and let $R_A, R_B$ be the radii of $A$, $B$ respectively. Define $A_\epsilon$ to be the subset of set $A$ that is less than $D(1 + \epsilon)$ distance from $B$, and define $B_\epsilon$ similarly. Consider two batches of size $k$ drawn from $p$ and matched with optimal transport. Then, for $k$ large enough, with high probability all cross-cluster matches will have an endpoint each in $A_\epsilon$ and $B_\epsilon$, where $\epsilon = \frac{\max(R_A, R_B)^2}{D^2}$.*

Theorem 2 implies for large $k$ that the vicinal distribution created by sampling along the cross-cluster matches will almost entirely lie in the voids between the clusters. If the clusters correspond to different classes, this will directly encourage the learned model to smoothly interpolate between the class labels as one transitions across the void between clusters. This is in contrast to the randomized matches of 1-mixup, which create vicinal distributions that can span a line crossing any part of the space without regard for intervening clusters. The behavior noted by Theorems 1 and 2 are visualized in Figures 1 and 2, showing that $k$-mixup provides smooth interpolation between clusters, and strengthens label preservation within clusters.

## 3 REGULARIZATION EXPANSIONS

Two recent works analyze the efficacy of 1-mixup perturbatively (Zhang et al., 2020; Carratino et al., 2020). Both consider quadratic Taylor series expansions about the training set or a simple transformation of it, and they characterize the regularization terms that arise in terms of label and Lipschitz smoothing. We adapt these expansions to $k$-mixup and show that the resulting regularization is more locally informed via the optimal transport coupling.

In both works, perturbations are sampled from a globally informed distribution, based upon all other samples in the training distribution. In $k$-mixup, these distributions are defined by the optimal transport couplings. Given a training point $x_i$, we consider all $k$-samplings $\gamma$ that might contain it, and all possible $k$-samplings $\xi$ that it may couple to. A locally-informed distribution is the following:

$$\mathcal{D}_i := \frac{1}{\binom{N-1}{k-1}\binom{N}{k}} \sum_{\gamma=1}^{\binom{N-1}{k-1}} \sum_{\xi=1}^{\binom{N}{k}} \delta_{\sigma_{\gamma\xi}(x_i)}$$

where $\sigma_{\gamma\xi}$ denotes the optimal coupling between $k$-samplings $\gamma$ and $\xi$. This distribution will be more heavily weighted on points that $x_i$ is often matched with. We use "locally-informed" in this sense of upweighting of points closer to the point of interest that are likely to be matched to it by $k$-mixup.

Zhang et al. (2020) expand about the features in the training dataset $\mathcal{D}_X := \{x_1, \ldots, x_n\}$, and the perturbations in the regularization terms are sampled from $\mathcal{D}$. We generalize their characterization to $k$-mixup, with $\mathcal{D}$ replaced by $\mathcal{D}_i$. We assume a loss of the form $\ell(f(x), y) = h(f(x)) - y \cdot f(x)$ for some twice differentiable $h$ and $f$. This broad class of losses includes the cross-entropy for neural networks and all losses arising from Generalized Linear Models.

**Theorem 3.** *Assuming a loss $\ell$ as above, the $k$-mixup loss can be written as:*

$$\mathcal{E}_k^{mix}(f) = \mathcal{E}^{std} + \sum_{j=1}^{3} \mathcal{R}_j + \mathbb{E}_{\lambda\sim\beta(\alpha+1,\alpha)}[(1-\lambda)^2\phi(1-\lambda)]$$

*where $\lim_{a\to 0}\phi(a) = 0$, $\mathcal{E}^{std}$ denotes the standard ERM loss, and the three $\mathcal{R}_i$ regularization terms are:*

$$\mathcal{R}_1 = \frac{\mathbb{E}_{\lambda\sim\beta(\alpha+1,\alpha)}[1-\lambda]}{n} \sum_{i=1}^{N} (h'(f(x_i)) - y_i)\nabla f(x_i)^T \mathbb{E}_{r\sim\mathcal{D}_i}[r - x_i]$$

$$\mathcal{R}_2 = \frac{\mathbb{E}_{\lambda\sim\beta(\alpha+1,\alpha)}[(1-\lambda)^2]}{2n} \sum_{i=1}^{N} h''(f(x_i))\nabla f(x_i)^T \mathbb{E}_{r\sim\mathcal{D}_i}[(r - x_i)(r - x_i)^T]\nabla f(x_i)$$

$$\mathcal{R}_3 = \frac{\mathbb{E}_{\lambda\sim\beta(\alpha+1,\alpha)}[(1-\lambda)^2]}{2n} \sum_{i=1}^{N} (h'(f(x_i)) - y_i)\mathbb{E}_{r\sim\mathcal{D}_i}[(r - x_i)\nabla^2 f(x_i)(r - x_i)^T].$$

A proof is given in Section F of the supplement and follows from some algebraic rearrangement and a Taylor expansion in terms of $1 - \lambda$. The higher-order terms are captured by $\mathbb{E}_{\lambda\sim\beta(\alpha+1,\alpha)}[(1-\lambda)^2\phi(1-\lambda)]$. $\mathcal{E}^{std}$ represents the constant term in this expansion, while the regularization terms $\mathcal{R}_i$ represent the linear and quadratic terms. These effectively regularize $\nabla f(x_i)$ and $\nabla^2 f(x_i)$ with respect to local perturbations $r - x_i$ sampled from $\mathcal{D}_i$, ensuring that our regularizations are locally-informed. In other words, the regularization terms vary over the support of the dataset, at each point   NEW penalizing the characteristics of the locally-informed distribution rather than a global distribution. This allows the regularization to adapt better to local data (e.g. manifold) structure. For example, $\mathcal{R}_2$ and $\mathcal{R}_3$ penalize having large gradients and Hessian respectively along the directions of significant variance of the distribution $\mathcal{D}_i$ of points $x_i$ is likely to be matched to. When as $k = 1$, this $\mathcal{D}_i$ will

not be locally informed, and will instead effectively be a global variance measure. As $k$ increases, the $\mathcal{D}_i$ will instead be dominated by matches to nearby clusters, better capturing the smoothing needs in the immediate vicinity of $x_i$. Notably, the expansion is in the feature space alone, yielding theoretical results in the case of 1-mixup on generalization and adversarial robustness.

An alternate approach by Carratino et al. (2020) characterizes mixup as a combination of a reversion to mean followed by random perturbation. In supplement Section G we generalize their result to $k$-mixup via a locally-informed mean and covariance.

## 4 EMPIRICAL RESULTS

We empirically test the efficacy of $k$-mixup with toy datasets, UCI datasets, image, and speech datasets employing a variety of neural network architectures. Across these experiments we find that $k$-mixup for $k > 1$ tends to improve upon (1-)mixup for fixed $\alpha$, and if not remains comparable. Additional experiments with CIFAR-10 compare $k$-mixup with manifold mixup, explore the combination of the two methods, and demonstrate improvements in adversarial robustness from $k$-mixup. Experiments were performed in various laptop, cluster, and cloud environments using PyTorch. Unless otherwise stated, a standard SGD optimizer was used, with learning rate 0.1 decreased at epochs 100 and 150, momentum 0.9, and weight decay $10^{-4}$.

**Toy datasets.** Results for the toy datasets of Figures 1 and 2 (denoted "One Ring," "Four Bars," and "Swiss Roll") are shown in Figure 3. We used a fully-connected 3-layer neural network (130 and 120 hidden units). As the datasets are very clustered and have no noise, smoothing is not needed for generalization and performance without mixup is typically 100%. Applying mixup to these datasets thus provides a view to the propensity of each variant to oversmooth, damaging performance. For each dataset and each $\alpha$, higher $k$-mixup outperforms (1-)mixup. For larger $\alpha$, the performance gap between the baseline (1-)mixup and $k > 1$ mixup becomes quite significant, reaching 8%, 50%, and 40% respectively for $k = 16$ and $\alpha = 64$. These results quantitatively confirm the intuition built in Figures 1 and 2 that $k$-mixup regularization more effectively preserves these structures in data, limiting losses from oversmoothing.

NEW

| $k$ | $\alpha=.25$ | $\alpha=1$ | $\alpha=4$ | $\alpha=16$ | $\alpha=64$ | $\alpha=.25$ | $\alpha=1$ | $\alpha=4$ | $\alpha=16$ | $\alpha=64$ | $\alpha=.25$ | $\alpha=1$ | $\alpha=4$ | $\alpha=16$ | $\alpha=64$ |
|---|---|---|---|---|---|---|---|---|---|---|---|---|---|---|---|
| 1 | 94.487 | 90.930 | 86.527 | 86.467 | 86.899 | 100 | 100 | 100 | 50.110 | 50.017 | 99.708 | 99.670 | 67.942 | 61.633 | 59.573 |
| 2 | 95.230 | 93.197 | 90.653 | 90.043 | 90.027 | 100 | 100 | 100 | 60.725 | 50.107 | 99.653 | 99.667 | 81.696 | 69.327 | 64.397 |
| 4 | 96.190 | 94.110 | 92.633 | 92.191 | 91.807 | 100 | 100 | 100 | 98.750 | 92.477 | 99.703 | **99.717** | **99.721** | 80.064 | 73.773 |
| 8 | 96.547 | 95.714 | 94.377 | 93.807 | 93.463 | 100 | 100 | 100 | 99.920 | 99.853 | 99.693 | 99.69 | 99.646 | 99.710 | 98.897 |
| 16 | **96.717** | **95.961** | **95.097** | **95.017** | **95.193** | 100 | 100 | 100 | **99.997** | **99.987** | **99.760** | 99.707 | 99.692 | **99.787** | **99.761** |
| | (a) One Ring | | | | | (b) Four Bars | | | | | (c) Swiss Roll | | | | |

Figure 3: Test accuracy on toy datasets, averaged over 5 Monte Carlo trials.

**UCI datasets.** Scaling up from the toy datasets, we tried $k$-mixup on UCI datasets (Dua & Graff, 2017) of varying size and dimension: Iris (150 instances, dim. 4), Breast Cancer Wisconsin-Diagnostic (569 instances, dim. 30), Abalone (4177 instances, dim. 8), Arrhythmia (452 instances, dim. 279), HTRU2 (17898 instances, dim. 9), and Phishing (11055 instances, dim. 30). For Iris, we used a 3-layer network with 120 and 84 hidden units; for Breast Cancer, Abalone, and Phishing, we used a 4-layer network with 120, 120, and 84 hidden units; for Arrhythmia we used a 5-layer network with 120, 120, 36, and 84 hidden units; and lastly, for HTRU2, we used a 5 layer network with 120, 120, 84, 132 hidden units. Each entry is averaged over 20 Monte Carlo trials. Test classification performance is shown in Figure 4. $k$-mixup improves over (1-)mixup in each case, although for Arrhythmia no mixup outperforms both, and for Phishing, no mixup statistically matches the performance of $k$-mixup. In these small datasets, the best (or at least good) performance seems to be achieved with relatively small $\alpha = 0.1$ and moderate $k$ (4 or 8).

| Data Set | None | $k=1$ | $k=2$ | $k=4$ | $k=8$ | $k=16$ | $k=1$ | $k=2$ | $k=4$ | $k=8$ | $k=16$ | $k=1$ | $k=2$ | $k=4$ | $k=8$ | $k=16$ |
|---|---|---|---|---|---|---|---|---|---|---|---|---|---|---|---|---|
| Abalone | 27.68 | 28.00 | 27.99 | 28.12 | 27.93 | 27.75 | 27.50 | 27.51 | 27.43 | **28.59** | 28.12 | 27.08 | 27.47 | 27.38 | 27.66 | 27.93 |
| Arrhythmia | **57.07** | 52.12 | 56.41 | 53.86 | 54.73 | 54.73 | 53.04 | 55.16 | 52.12 | 55.11 | 54.40 | 55.05 | 53.37 | 53.70 | 54.46 | 56.58 |
| Cancer | 92.81 | 92.10 | 92.91 | **93.79** | 91.95 | 91.99 | 91.76 | 90.82 | 89.96 | 89.26 | 87.70 | 91.72 | 92.27 | 92.32 | 91.58 | 91.10 |
| HTRU2 | 97.50 | 97.47 | **97.65** | 97.48 | 97.52 | 97.49 | 97.37 | 97.45 | 97.51 | 97.38 | 97.42 | 97.51 | 97.52 | 97.42 | 97.51 | 97.42 |
| Iris | 95.9 | 96.2 | 94.8 | 96.4 | **96.9** | 96.5 | 91.9 | 92 | 88.8 | 89.1 | 86 | 87.4 | 81 | 79.5 | 77.1 | 76.6 |
| Phishing | **96.57** | 96.26 | 96.39 | 96.44 | **96.56** | 96.30 | 95.89 | 95.89 | 95.99 | 96.01 | 96.33 | 94.73 | 95.06 | 94.90 | 95.56 | 96.00 |
| | | (a) $\alpha = 0.1$ | | | | | (b) $\alpha = 1.0$ | | | | | (c) $\alpha = 10.0$ | | | | |

Figure 4: Test accuracy on UCI datasets using fully connected networks.

| $k$ | $\alpha=.05$ | $\alpha=.1$ | $\alpha=.2$ | $\alpha=.5$ | $\alpha=1$ | $\alpha=10$ | $\alpha=100$ |
|---|---|---|---|---|---|---|---|
| 1 | 99.09 | 99.09 | 99.05 | 99.03 | 98.98 | 98.79 | 98.64 |
| 2 | 99.11 | 99.12 | 99.07 | 99.02 | 98.99 | 98.88 | 98.75 |
| 4 | 99.13 | 99.11 | 99.09 | 99.06 | 99.04 | 98.91 | 98.85 |
| 8 | 99.12 | 99.12 | 99.09 | 99.06 | 99.01 | 99.00 | 98.99 |
| 16 | 99.18 | 99.15 | 99.14 | 99.08 | 99.04 | 99.08 | 99.12 |
| 32 | 99.18 | 99.15 | 99.17 | 99.10 | 99.11 | 99.16 | **99.18** |

(a) Performance (test accuracy)

| $k$ | $\alpha=.05$ | $\alpha=.1$ | $\alpha=.2$ | $\alpha=.5$ | $\alpha=1$ | $\alpha=10$ | $\alpha=100$ |
|---|---|---|---|---|---|---|---|
| 1 | 0.068 | 0.148 | 0.239 | 0.436 | 0.617 | 1.347 | 1.732 |
| 2 | 0.070 | 0.129 | 0.214 | 0.388 | 0.586 | 1.228 | 1.585 |
| 4 | 0.049 | 0.111 | 0.191 | 0.338 | 0.542 | 1.078 | 1.399 |
| 8 | 0.037 | 0.092 | 0.149 | 0.305 | 0.449 | 0.930 | 1.191 |
| 16 | 0.035 | 0.070 | 0.122 | 0.241 | 0.341 | 0.719 | 0.938 |
| 32 | 0.029 | 0.039 | 0.076 | 0.152 | 0.221 | 0.456 | 0.586 |

(b) Average squared distance of vicinal distribution from training set

Figure 5: Results for MNIST with a LeNet architecture (no mixup performance: 99.0%), averaged over 20 Monte Carlo trials ($\pm.02$ confidence on test performance). Note that $k$-mixup doubles the improvement of 1-mixup over ERM.

| $k$ | $\alpha=.05$ | $\alpha=.1$ | $\alpha=.2$ | $\alpha=.5$ | $\alpha=1$ | $\alpha=10$ | $\alpha=100$ |
|---|---|---|---|---|---|---|---|
| 1 | 94.785 | 95.025 | 95.27 | 95.645 | 95.63 | 94.82 | 94.085 |
| 2 | 94.8 | 94.92 | 95.285 | 95.645 | 95.79 | 95.105 | 94.14 |
| 4 | 94.76 | 94.99 | 95.3 | 95.65 | 95.745 | 95.205 | 94.315 |
| 8 | 94.79 | 94.925 | 95.255 | 95.625 | **95.815** | 95.285 | 94.61 |
| 16 | 94.68 | 94.98 | 95.215 | 95.595 | 95.735 | 95.33 | 94.92 |
| 32 | 94.74 | 94.905 | 95.11 | 95.465 | 95.675 | 95.375 | 95.165 |

(a) Performance (test accuracy)

| $k$ | $\alpha=.05$ | $\alpha=.1$ | $\alpha=.2$ | $\alpha=.5$ | $\alpha=1$ | $\alpha=10$ | $\alpha=100$ |
|---|---|---|---|---|---|---|---|
| 1 | 29.5 | 58.2 | 107.1 | 198.1 | 280.3 | 624.1 | 803.1 |
| 2 | 27.0 | 54.0 | 94.4 | 192.6 | 276.8 | 564.9 | 721.2 |
| 4 | 22.3 | 58.6 | 87.3 | 163.0 | 240.6 | 494.3 | 637.6 |
| 8 | 24.1 | 43.1 | 78.4 | 141.7 | 199.5 | 431.3 | 554.8 |
| 16 | 16.8 | 32.6 | 70.7 | 114.3 | 186.6 | 365.2 | 468.7 |
| 32 | 14.9 | 28.4 | 50.4 | 96.9 | 142.4 | 288.6 | 371.7 |

(b) Average squared distance of vicinal distribution from training set.

Figure 6: Results for CIFAR-10 with Resnet18 architecture (no mixup performance: 94.4%), averaged over 20 Monte Carlo trials ($\pm.03$ confidence on test performance). Difference between best $k$-mixup and best 1-mixup is 0.17% for fixed high $\alpha$ ($\alpha=100$), the improvement increases to 1.2%.

**Image datasets.** Results for MNIST (LeCun & Cortes, 2010) using a convolutional neural network, i.e., LeNet modified slightly to accommodate grayscale images, are in Figure 5, with a joint sweep over $k$ and $\alpha$. Each table entry is averaged over 20 Monte Carlo trials. For each generated point in the vicinal distribution, we compute its closest squared distance to the matched pair from which it was generated. The results are averaged over the vicinal dataset and reported in part (b) of the figure. It shows that $k$-mixup yields significantly closer matches as $k$ increases, causing the generated vicinal distribution to deviate less (in terms of squared distance) from the original training set. This is empirical evidence of our method respecting manifold support and cluster structure, as noted in Section 2. In spite of this lower variation, for each $\alpha$ the best generalization performance is for some $k > 1$, with $\alpha = 100$ yielding 99.18% accuracy for $k = 32$ compared to 98.64% accuracy for $k = 1$. Generally, for fixed $\alpha$, increasing $k$ improves performance for this dataset.

Figure 6 shows analogous results for CIFAR-10, using the PreAct Resnet-18 architecture as in Zhang et al. (2018). Again $k$-mixup succeeds in finding matches that lie significantly closer together as $k$ increases. For each $\alpha$ except the small $\alpha = .1$, the best generalization performance is still for some $k > 1$, with $\alpha = 100$ yielding 95.165% accuracy for $k = 32$ compared to 94.085% accuracy for $k = 1$. The best performance overall is achieved at $k = 8$ with parameter $\alpha = 1$. While the best $k$-mixup performance exceeds that of the best 1-mixup by only 0.17%, recall that in this setting 1-mixup outperforms ERM by 1.4% (Zhang et al., 2018), so when combined with the low overall error rate, small gains are not surprising. Tables of results for DenseNet and WideResnet architectures can be found in Figure 9, with the best $k$-mixup outperforming the best 1-mixup by 0.44% and 0.28% respectively. Note that in the case of Densenet, $k = 16$ outperforms or (statistically) matches $k = 1$ for all values of $\alpha$.

| $k$ | $\alpha=.1$ | $\alpha=.2$ | $\alpha=.5$ | $\alpha=1$ | $\alpha=10$ |
|---|---|---|---|---|---|
| 1 | 76.54 | 77.22 | 77.88 | 77.76 | 74.88 |
| 2 | 76.54 | 77.38 | 78.19 | **78.41** | 76.09 |
| 4 | 76.55 | 77.36 | 77.94 | 78.25 | 76.51 |
| 8 | 76.51 | 77.18 | 77.82 | 78.03 | 76.69 |
| 16 | 76.43 | 77.08 | 77.77 | 77.87 | 76.86 |
| 32 | 76.29 | 76.93 | 77.51 | 77.54 | 76.90 |

CIFAR-100 performance ($\pm.05$ confidence)

| $k$ | $\alpha=.1$ | $\alpha=.2$ | $\alpha=.5$ | $\alpha=1$ | $\alpha=10$ |
|---|---|---|---|---|---|
| 1 | 97.13 | 97.12 | 97.14 | 97.17 | 96.76 |
| 2 | 97.16 | 97.14 | 97.19 | **97.29** | 96.81 |
| 4 | 97.11 | 97.13 | 97.16 | 97.24 | 96.87 |
| 8 | 97.16 | 97.11 | 97.14 | 97.23 | 96.94 |
| 16 | 97.09 | 97.11 | 97.08 | 97.16 | 96.93 |
| 32 | 97.16 | 97.10 | 97.05 | 97.08 | 96.94 |

(b) SVHN performance ($\pm.02$ confidence)

Figure 8: Test accuracy on CIFAR-100 and SVHN, averaged over 20 Monte Carlo trials ($\pm.03$ confidence).

For both MNIST and CIFAR-10 datasets, with coupling distances reported, one could compare settings with similar average squared distances. For example, a close inspection of Figures 5 and 6 show that higher $k$ again tends to have the advantage, indicating that performance gains are not simply due to having a closer vicinal distribution. NEW

Additionally, we show training curves (performance as a function of epoch) for Resnet-18 on CIFAR-10 in Figure 7, averaged over 20 random trials. The training speed (test accuracy in terms of epoch) for 1-mixup and 32-mixup shows no loss of convergence speed with $k = 32$ mixup, with (if anything) $k = 32$ showing a slight edge. The discontinuity at epoch 100 is due to our reduction of the learning

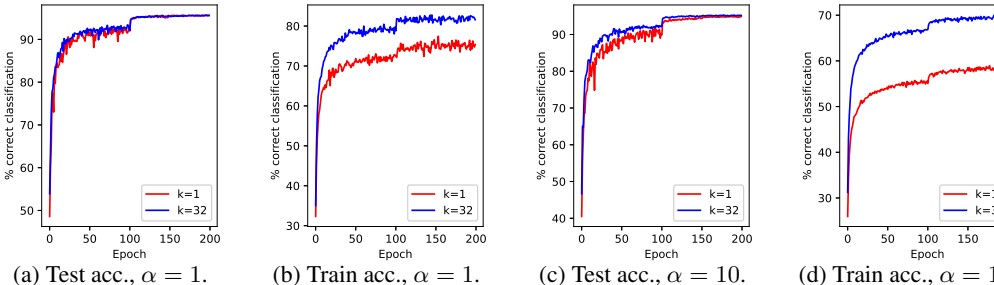

(a) Test acc., $\alpha = 1$.    (b) Train acc., $\alpha = 1$.    (c) Test acc., $\alpha = 10$.    (d) Train acc., $\alpha = 10$.

Figure 7: Training convergence of $k = 1$ and $k = 32$ mixup on CIFAR-10, averaged over 20 random trials. Note that both train at roughly the same rate ($k = 32$ slightly faster), the train accuracy discrepancy is due to the more class-accurate vicinal training distribution created by higher $k$-mixup.

| $k$ | $\alpha = .5$ | $\alpha = 1$ | $\alpha = 2$ | $\alpha = 4$ |
|---|---|---|---|---|
| 1 | 96.65 | 96.71 | 96.58 | 96.43 |
| 16 | 96.62 | 97.02 | **97.15** | 97.08 |

(a) DenseNet-BC-190 architecture
($\pm.03$ confidence, no mixup performance: 96.3%)

| $k$ | $\alpha = .05$ | $\alpha = .2$ | $\alpha = .5$ | $\alpha = 1$ | $\alpha = 2$ | $\alpha = 4$ |
|---|---|---|---|---|---|---|
| 1 | 88.46 | 88.47 | 88.38 | 88.22 | 87.75 | 87.01 |
| 4 | 88.64 | 88.53 | 88.73 | 88.59 | 88.41 | 87.84 |
| 16 | 88.47 | 88.41 | **88.75** | 88.66 | 88.62 | 88.26 |

(b) WideResnet-101 architecture
($\pm.09$ confidence, no mixup performance: 88.4%)

Figure 9: CIFAR-10 test accuracy for DenseNet-BC-190 and WideResnet-101 architectures. For DenseNet, the difference between best $k$-mixup and best 1-mixup is 0.44%, for fixed high $\alpha$ ($\alpha = 4$), the improvement increases to 0.65%. For WideResnet, the difference between best $k$-mixup and best 1-mixup is 0.28%, for fixed high $\alpha$ ($\alpha = 4$), the improvement increases to 1.25%.

rate at epochs 100 and 150 to aid convergence (used throughout our image experiments). The train accuracy shows similar convergence profile between 1- and 32-mixup; the difference in absolute accuracy here (and the reason it is less than the test accuracy) is because the training distribution is the mixup-modified vicinal distribution. The curve for $k = 32$ is higher, especially for $\alpha = 10$, because the induced vicinal distribution and labels are more consistent with the true distribution, due to the better matches from optimal transport. The large improvement in train accuracy is remarkable given the high dimension of the CIFAR-10 data space, since it indicates that $k = 32$ mixup is able to find significantly more consistent matches than $k = 1$ mixup.

Figure 8 shows results for CIFAR-100 and SVHN, with a Resnet-18 architecture. As before, for fixed $\alpha$, the best performance is achieved for some $k > 1$. The improvement of the best $k$-mixup over the best 1-mixup is 0.53% for CIFAR-100 and 0.12% for SVHN. For fixed high $\alpha = 10$, the $k$-mixup improvement over 1-mixup rises to 2.02% for CIFAR-100, possibly indicating that the OT matches yield better interpolation between classes, aiding generalization.

**Speech dataset**. Performance is tested on a speech dataset: Google Speech Commands (Warden, 2018) using a LeNet architecture are in Figure 10. Each table entry is averaged over 20 Monte Carlo trials ($\pm.014$ confidence on test performance). We augmented the data in the same way as Zhang et al. (2018). The difference between best $k$-mixup and best 1-mixup is 0.4%.

| $k$ | $\alpha = .1$ | $\alpha = .2$ | $\alpha = .5$ | $\alpha = 1$ | $\alpha = 10$ |
|---|---|---|---|---|---|
| 1 | 89.05 | 89.03 | 89.40 | 89.42 | 88.28 |
| 2 | 89.17 | 89.12 | 89.40 | 89.43 | 88.67 |
| 4 | 89.07 | 89.06 | **89.82** | 89.61 | 89.13 |
| 8 | 88.98 | 89.05 | 89.73 | 89.50 | 89.49 |
| 16 | 88.97 | 88.96 | 89.58 | 89.68 | 89.74 |

Figure 10: Google Speech Commands test accuracy using LeNet architecture, averaged over 20 Monte Carlo trials ($\pm.014$ confidence).

**Manifold mixup.** We compare to Manifold Mixup (Verma et al., 2018), which helps in settings like those in Figure 1 by making interpolations more meaningful. It performs mixup at random layers (random per minibatch), not only in the input space. This extends to "manifold $k$-mixup:" $k$-mixup in the hidden layers as well as the input layers. We use settings in Verma et al. (2018), i.e., for PreAct Resnet18, the mixup layer is randomized (coin flip) between the input space and the output of the first residual block.[2]

Results for CIFAR-10 with a Resnet18 architecture are in Figure 11. Numbers in this experiment are averaged over 20 Monte Carlo trials ($\pm.03$ confidence on test performance). While manifold 1-mixup outperforms the standard 1-mixup from Figure 6, it is matched by manifold $k$-mixups with $k = 4$ and outperformed by standard $k$-mixup (Figure 6, $k = 8$). We also tried randomizing over (a) the

---

[2]See `https://github.com/vikasverma1077/manifold_mixup`

outputs of all residual blocks and (b) the outputs of (lower-dimensional) deep residual blocks only, but found that performance of both 1-mixup and $k$-mixup degrades in these cases. This underscores that mixup in hidden layer manifolds is not guaranteed to be effective and can require tuning.

**Adversarial robustness.** Figure 12 shows results on white-box adversarial attacks generated by the FGSM method (implementation of Kim (2020)[3]) for various values of maximum adversarial perturbation. As in Verma et al. (2018), we used FGSM over PGD, since the iterative PGD attacks are significantly more effective, making any performance improvements seem less relevant in practice. Additionally, we expect simple gradient attacks to be particularly meaningful for evaluating the success of mixup,

| $k$ | $\alpha = .1$ | $\alpha = .2$ | $\alpha = .5$ | $\alpha = 1$ | $\alpha = 10$ |
|---|---|---|---|---|---|
| 1 | 95.00 | 95.27 | 95.59 | 95.74 | 95.19 |
| 2 | 94.91 | 95.22 | 95.56 | 95.69 | 95.29 |
| 4 | 94.99 | 95.22 | 95.55 | **95.75** | 94.98 |
| 8 | 94.92 | 95.22 | 95.58 | 95.72 | 95.39 |
| 16 | 94.86 | 95.18 | 95.58 | 95.69 | 95.47 |
| 32 | 94.86 | 95.09 | 95.42 | 95.53 | 95.46 |

Figure 11: Manifold mixup test accuracy on CIFAR-10 using Resnet18 architecture, averaged over 20 Monte Carlo trials ($\pm.03$ confidence).

whose goal is to approximately linearly interpolate between classes. We show CIFAR-10 accuracy on white-box FGSM adversarial data (10000 points), where the maximum adversarial perturbation is set to $\epsilon/255$; performance is averaged over 30 Monte Carlo trials ($\pm0.6$ confidence). All performances are without adversarial training. Note that $k = 2$ outperforms $k = 1$ uniformly by as much as 8.64%, and the $k > 1$ mixups all outperform $k = 1$ mixup on larger attacks. Similar results for MNIST are in Figure 12(b), with the FGSM attacks being somewhat less effective. Here $k = 8$ is best for smaller attacks, and $k = 2$ is best for larger attacks.

The improved robustness shown by $k$-mixup speaks to a key goal of mixup, that of smoothing the predictions in the parts of the data space where no/few labels are available. This smoothness should make adversarial attacks require greater magnitude to successfully "break" the model.

| $k$ | $\epsilon = .5$ | $\epsilon = 1$ | $\epsilon = 2$ | $\epsilon = 4$ | $\epsilon = 8$ | $\epsilon = 16$ |
|---|---|---|---|---|---|---|
| 1 | 78.46 | 72.07 | 66.02 | 59.72 | 48.61 | 23.71 |
| 2 | **79.48** | **74.22** | **70.18** | **66.58** | **57.25** | 26.23 |
| 4 | 78.82 | 72.71 | 67.82 | 63.60 | 54.80 | **27.32** |
| 8 | 78.00 | 70.45 | 64.24 | 59.01 | 50.32 | 26.24 |

(a) CIFAR-10 ($\pm0.6$ confidence)

| $k$ | $\epsilon = .5$ | $\epsilon = 1$ | $\epsilon = 2$ | $\epsilon = 4$ | $\epsilon = 8$ | $\epsilon = 16$ |
|---|---|---|---|---|---|---|
| 1 | 98.26 | 97.36 | 96.03 | 92.97 | 86.89 | 77.14 |
| 2 | 98.32 | 97.45 | **96.15** | **93.16** | **87.34** | **78.24** |
| 4 | 98.30 | 97.30 | 95.81 | 92.34 | 84.89 | 72.48 |
| 8 | **98.36** | **97.46** | 95.96 | 92.33 | 83.70 | 69.33 |

(b) MNIST ($\pm0.1$ confidence)

Figure 12: Adversarial robustness: accuracy on white-box FGSM adversarial attacks.

## 5 CONCLUSIONS AND FUTURE WORK

The experiments above demonstrate that $k$-mixup often improves the generalization and robustness gains achieved by (1-)mixup, and at least does no harm. This is seen across a diverse range of datasets and network architectures. It is simple to implement, adds little computational overhead to conventional (1-)mixup training, and may also be combined with related mixup variants. As $k$ increases, the regularization induced more accurately reflects the local structure of the training data, especially in the manifold support and clustered settings, as seen in Sections 2 and 3. Empirical results show that performance is robust to variations in $k$, ensuring that extensive tuning is not required.

Our experiments show the greatest improvement from using $k$-mixup on low-dimensional datasets, while the improvement on high-dimensional datasets remains positive but is smaller (recall that classic mixup also has somewhat small gains over no mixup in these settings). This difference could be due to the diminishing value of Euclidean distance for characterizing dataset geometry in high dimensions (Aggarwal et al., 2001), but intriguingly this effect was not remedied by doing OT in the possibly lower-dimensional manifolds created by the higher layers in our manifold mixup experiments. To remedy this issue, in future work we will consider alternative metric learning strategies, with the goal of identifying effective high-dimensional metrics for displacement interpolation of data points.

Mixup has been incorporated into a broad range of learning applications, many of which we have not empirically investigated here. The original work demonstrated robustness to label corruption and improved GAN training stability, which could easily benefit from $k$-mixup as well. Additionally, mixup has been used for augmentation in semi-supervised learning (Berthelot et al., 2019; Sohn et al., 2020b). Given its simplicity, a $k$-mixup augmentation may be able to generate more informed shifts of labelled and unlabelled training points. This same philosophy applies to a recent method for generalizing fair classifiers (Chuang & Mroueh, 2021), which already leverages manifold mixup.

---

[3]Software has MIT License

**Ethics Statement** The $k$-mixup regularization procedure we propose and our experimentation on it do not bring up additional ethical concerns outside of general concerns associated with machine learning as a field, and the use of neural network models. Standard benchmark datasets and tasks were used, with the assumption that dataset originators have done their due diligence with respect to concerns on IRBs, privacy, fairness, and the like.

**Reproducibility Statement** With regards to reproducibility, we note that $k$-mixup is exceedingly simple, and may be easily incorporated into almost any neural network as a data perturbation step. For our experiments, publicly available benchmark datasets have been used and are clearly stated in Section 4. Additionally, the models used and the relevant training parameters are described for each task considered. Upon acceptance, we would be glad to make a full implementation of our experiments available on Github under an open source MIT license. Lastly, for theoretical results, full proofs of our statements are provided in the appendices following the references.

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

## A    $k$-MIXUP PSEUDOCODE & COMPUTATIONAL COST

NEW

Below is code for one epoch of $k$-mixup training, in the style of Figure 1 of Zhang et al. (2018).

```
# y1, y2 should be one-hot vectors
for (x1, y1), (x2, y2) in zip(loader1, loader2):
    cost = numpy.inf * [x1.shape[0], x1.shape[0]]
    for i in range(x1.shape[0] // k):
        cost[i * k:(i+1) * k, i * k:(i+1) * k] = scipy.spatial.
            distance_matrix(x1[i * k:(i+1) * k], x2[i * k:(i+1) * k])
        _, idx = scipy.optimize.linear_sum_assignment(cost)
        x2[i * k:(i+1) * k] = x2[idx, i * k:(i+1) * k]
        y2[i * k:(i+1) * k] = y2[idx, i * k:(i+1) * k]
    lam = numpy.random.beta(alpha, alpha)
    x = Variable(lam * x1 + (1 - lam) * x2)
    y = Variable(lam * y1 + (1 - lam) * y2)
    optimizer.zero_grad()
    loss(net(x), y).backward()
    optimizer.step()
```

**Computational cost.** While the cost of the Hungarian algorithm is $O(k^3)$, it provides $k$ data points for regularization, yielding an amortized $O(k^2)$ complexity per data point. For the smaller values of $k$ that we empirically consider, approximate Sinkhorn-based methods (Cuturi, 2013) are slower in practice, and the Hungarian cost remains small relative to that of gradient computation (e.g. for CIFAR-10 and $k = 32$, the Hungarian algorithm costs 0.69 seconds per epoch in total). Computing the distance matrix input to the OT matching costs $O(k^2 d)$ where $d$ is dimension, yielding an amortized $O(kd)$ complexity per data point. With the high dimensionality of CIFAR, a naive CPU implementation of this step of the computation adds about 8 seconds per epoch.[4] Moreover, training convergence speed is unaffected, unlike manifold mixup (Verma et al., 2018), which in our experience converges slower and has larger computational overhead. Overall, there is little computational downside to generalizing 1-mixup to $k$-mixup.

## B    PROOF OF PROPOSITION 1

NEW

Finite-sample convergence of empirical measures implies that for an arbitrary sampling of $k$ points $\hat{\mu}_k$, we have $W_2^2(\mu, \hat{\mu}_k) \leq O(k^{-2/d})$ with high probability (e.g., $1 - 1/k^2$ probability, from Theorem 9.1 of Solomon et al. (2020) when combined with results from Weed & Bach (2019)). The triangle inequality then implies that the Wasserstein-2 distance between our batches of $k$ samples will tend to 0 at the same asymptotic rate: $W_2^2(\hat{\mu}_k^\gamma, \hat{\mu}_k^\xi) \leq O(k^{-2/d})$ with high probability.

Recalling that by the definition of the optimal coupling permutation $\sigma(i)$, we have $\frac{1}{k} \sum_{i=1}^{k} \|x_i^\gamma - x_{\sigma(i)}^\xi\|_2^2 = W_2^2(\hat{\mu}_k^\gamma, \hat{\mu}_k^\xi) \leq O(k^{-2/d})$ with high probability. Hence, for any $\mathcal{I} \subseteq [1, k]$ with $\|x_i^\gamma - x_{\sigma(i)}^\xi\|_2^2 > k^{-1/d}$ for all $i \in \mathcal{I}$, $|\mathcal{I}| \leq O(k^{1-1/d}) \leq \delta k$ for $k$ large enough and any $\delta \in (0, 1]$. In essence, long-distance identifications are rare, and their fraction is bounded for large enough $k$. Therefore, for $k$ large enough, again with high probability there exists a subset $\bar{\mathcal{I}}$ with $|\bar{\mathcal{I}}| \geq (1 - \delta)k$ such that $\|x_i^\gamma - x_{\sigma(i)}^\xi\|_2 \leq R_\mathcal{S}$ for all $i \in \bar{\mathcal{I}}$. Since $x_i^\gamma$ and $x_{\sigma(i)}^\xi$ lie in $\mathcal{S}$, the proposition results.

## C    PROOF OF LEMMA 1

We argue by contradiction and prove the result for $m = 2$ first. Suppose that the number of cross-cluster matchings exceeds $|r_1 - s_1|$. Then by the pigeonhole principle, there must be at least two such matchings, which we'll say are between $p_i$ and $q_i$, and $p_{i+1}$ and $q_{i+1}$, WLOG. As the cost of the cross-cluster matchings is $\geq (2\Delta)^2$, and the cost of intra-cluster matchings is $\leq (2\Delta)^2$, a more

---

[4]It is possible to accelerate this computation with parallelization/GPU usage, but as this cost is a small portion of the overall training cost we did not optimize it.

optimal matching would pair $p_i$ and $q_{i+1}$, and $p_{i+1}$ and $q_i$. This contradicts optimality of the initial pairing.

In the scenario with $m$ clusters, an analogous argument works. As above, $|r_i - s_i|$ is the number of cluster $i$ elements that must be matched in a cross-cluster fashion, and half the sum of these quantities is the minimum number of cross-cluster matchings overall. If this number is exceeded, by the pigeonhole principle, there must be additional cross-cluster matches that form a cycle in the graph over clusters. As above, the cost would be reduced by matching points within the same clusters, so this contradicts optimality.

## D    PROOF OF THEOREM 1

We first prove the result for two clusters of weight $p$ and $1 - p$. With our clustered assumption, we may assume that all identifications that can be intracluster will be intracluster. Thus, if $s_1$ and $s_2$ denote the number of points in cluster 1 from batch 1 and 2, then the resulting number of cross-cluster matches is $|s_1 - s_2|$. As the samples for our batches are i.i.d., these random variables follow a simple binomial distribution $B(k, p)$. We can bound the expectation of this quantity with Jensen's inequality:

$$(\mathbb{E}[|s_1 - s_2|])^2 \le \mathbb{E}[|s_1 - s_2|^2] = 2\text{Var}(s_1) = 2kp(1 - p)$$

With some algebraic manipulation, our desired rate bound results. It is also possible to get an exact rate with some hypergeometric identities (Katti, 1960), but these simply differ by a constant factor, so we omit the exact expressions here.

For the general case, our clustered assumption again allows us to assume that OT will prioritize intracluster identifications. Thus, if we let $r_i$ and $s_i$ denote the number of samples in cluster $i$ in batch 1 and 2, respectively, then the number of cross-cluster matchings will be $\frac{1}{2}\sum_i |r_i - s_i|$. Ultimately, $r_i$ and $s_i$ are sampled from a binomial distribution $B(k, p_i)$, so we may repeat the argument above and note that $\mathbb{E}[r_i = s_i] \le \sqrt{2kp_i(1 - p_i)}$. Simple algebraic manipulations lead to the desired result.

## E    PROOF OF THEOREM 2

By the smooth boundary and positive density assumptions, we know that $P(A_\delta) > 0$ and $P(B_\delta) > 0$ for any $\delta > 0$. Hence, for fixed $\delta$ and $k$ large enough, we know that with high probability the sets $A_\delta$ and $B_\delta$ each contain more points than the number of cross-cluster identifications.

Now consider $A_\epsilon$ and $B_\epsilon$ for $\epsilon = 2\delta + (\max(R_A, R_B)^2)/(2D^2)$. All cross-cluster matches need to be assigned. The cost of assigning a cross-cluster match to a point in $A_\delta$ and a point in $B_\delta$ is at most $D^2(1 + 2\delta)^2$ (since we are using $W_2$). Furthermore, the cost of assigning a cross-cluster match that contains a point in $A$ outside $A_\epsilon$ and an arbitrary point in $B$ is at least $D^2(1 + \epsilon)^2$. Consider the difference between these two costs:

$$D^2(1 + \epsilon)^2 - D^2(1 + 2\delta)^2 = D^2(2(\epsilon - 2\delta) + \epsilon^2 - 4\delta^2) > 2D^2 \frac{\max(R_A, R_B)^2}{2D^2} \ge R_A^2.$$

Since this difference $> 0$ and we have shown $A_\delta$ contains sufficient points for handling all assignments, this assignment outside of $A_\epsilon$ will only occur if there is a within-cluster pair which benefits from using the available point in $A_\epsilon$ more than is lost by not giving it to the cross-cluster pair ($> R_A^2$). The maximum possible benefit gained by the within-cluster pair is the squared radius of $A$, i.e. $R_A^2$. Since we have shown that the lost cost for the cross-cluster pair is bigger than $R_A^2$, we have arrived at a contradiction. The proof is similar for the $B$ side.

We have thus shown that for $k$ large enough (depending on $\delta$), with high probability all cross-cluster matches have an endpoint each in $A_\epsilon$ and $B_\epsilon$ where $\epsilon = 2\delta + (\max(R_A, R_B)^2)/(2D^2)$. Setting $\delta = (\max(R_A, R_B)^2)/(4D^2)$ completes the proof.

## F    PROOF OF THEOREM 3

We mostly follow the notation and argument of Zhang et al. (2020) (c.f. Lemma 3.1), modifying it for our setting. There they consider sampling $\lambda \sim Beta(\alpha, \beta)$ from an asymmetric Dirichlet

distribution. Here, we assume a symmetric Dirichlet distribution, such that $\alpha = \beta$, simplifying most of the expressions. The analogous results hold in the asymmetric case with simple modifications.

Consider the probability distribution $\tilde{\mathcal{D}}_\lambda$ with probability distribution: $\beta(\alpha + 1, \alpha)$. Note that this distribution is more heavily weighted towards 1 across all $\alpha$, and for $\alpha < 1$, there is an asymptote as you approach 1.

Let us adopt the shorthand notation $\tilde{x}_{i,\sigma_{\gamma\xi}(i)}(\lambda) := \lambda x_i^\gamma + (1-\lambda)x_{\sigma_{\gamma\xi}}^\xi$ for an interpolated feature point. The manipulations below are abbreviated, as they do not differ much for our generalization.

$$\mathcal{E}_k^{mix}(f) = \frac{1}{k\binom{N}{k}^2}\mathbb{E}_{\lambda\sim\beta(\alpha,\alpha)}\sum_{\gamma,\xi=1}^{\binom{N}{k}}\sum_{i=1}^k h(f(\tilde{x}_{i,\sigma_{\gamma\xi}(i)}(\lambda))) - (\lambda y_i^\gamma + (1-\lambda)y_{\sigma_{\gamma\xi}(i)}^\xi)f(\tilde{x}_{i,\sigma_{\gamma\xi}(i)}(\lambda))$$

$$= \frac{1}{k\binom{N}{k}^2}\mathbb{E}_{\lambda\sim\beta(\alpha,\alpha)}\mathbb{E}_{B\sim Bern(\lambda)}\sum_{\gamma,\xi=1}^{\binom{N}{k}}\sum_{i=1}^k B[h(f(\tilde{x}_{i,\sigma_{\gamma\xi}(i)}(\lambda))) - y_i^\gamma f(\tilde{x}_{i,\sigma_{\gamma\xi}(i)}(\lambda))]$$
$$+ (1-B)[h(f(\tilde{x}_{i,\sigma_{\gamma\xi}(i)}(\lambda))) - y_{\sigma_{\gamma\xi}(i)}^\xi f(\tilde{x}_{i,\sigma_{\gamma\xi}(i)}(\lambda))]$$

$$= \frac{1}{k\binom{N}{k}^2}\sum_{\gamma,\xi=1}^{\binom{N}{k}}\sum_{i=1}^k\mathbb{E}_{\lambda\sim\beta(\alpha+1,\alpha)}h(f(\tilde{x}_{i,\sigma_{\gamma\xi}(i)}(\lambda))) - y_i^\gamma f(\tilde{x}_{i,\sigma_{\gamma\xi}(i)}(\lambda))$$

For the third equality above, the ordering of sampling for $\lambda$ and $B$ has been swapped via conjugacy: $\lambda \sim \beta(\alpha, \alpha)$, $B|\lambda \sim Bern(\lambda)$ is equivalent to $B \sim \mathcal{U}\{0, 1\}$, $\lambda|B \sim \beta(\alpha + B, \alpha + 1 - B)$. This is combined with the fact that $\tilde{x}_{i,\sigma_{\gamma\xi}(i)}(1-\lambda) = \tilde{x}_{\sigma_{\gamma\xi}(i),i}(\lambda)$ to get the last line above.

Now we can swap the sums, grouping over the initial point to express this as the following:

$$\mathcal{E}_k^{mix}(f) = \frac{1}{N}\sum_{i=1}^N\mathbb{E}_{\lambda\sim\beta(\alpha+1,\alpha)}\mathbb{E}_{r\sim\mathcal{D}_i}h(f(\lambda x_i + (1-\lambda)r)) - y_i^\gamma f(\lambda x_i + (1-\lambda)r),$$

where the probability distribution $\mathcal{D}_i$ is as described in the text.

The remainder of the argument performs a Taylor expansion of the loss term $h(f(\lambda x_i + (1-\lambda)r)) - y_i^\gamma f(\lambda x_i + (1-\lambda)r)$ in terms of $1 - \lambda$, and is not specific to our setting, so we refer the reader to Appendix A.1 of (Zhang et al., 2020). for the argument.

## G $\quad$ $k$-MIXUP AS MEAN REVERSION FOLLOWED BY REGULARIZATION

**Theorem 4.** *Define $(\tilde{x}_i^\gamma, \tilde{y}_i^\gamma)$ as*
$$\tilde{x}_i^\gamma = \bar{x}_i^\gamma + \bar{\theta}(x_i^\gamma - \bar{x}_i^\gamma)$$
$$\tilde{y}_i^\gamma = \bar{y}_i^\gamma + \bar{\theta}(y_i^\gamma - \bar{y}_i^\gamma),$$
*where $\bar{x}_i^\gamma = \frac{1}{\binom{N}{k}}\sum_{\xi=1}^{\binom{N}{k}}x_{\sigma_{\gamma\xi}(i)}^\xi$ and $\bar{y}_i^\gamma = \frac{1}{\binom{N}{k}}\sum_{\xi=1}^{\binom{N}{k}}y_{\sigma_{\gamma\xi}(i)}^\xi$ are expectations under the matchings and $\theta \sim \beta_{[1/2,1]}(\alpha, \alpha)$. Further, denote the zero mean perturbations*
$$\tilde{\delta}_i^\gamma = (\theta - \bar{\theta})x_i^\gamma + (1-\theta)x_{\sigma_{\gamma\xi}(i)}^\xi - (1-\bar{\theta})\bar{x}_i^\gamma$$
$$\tilde{\epsilon}_i^\gamma = (\theta - \bar{\theta})y_i^\gamma + (1-\theta)y_{\sigma_{\gamma\xi}(i)}^\xi - (1-\bar{\theta})\bar{y}_i^\gamma.$$
*Then the $k$-mixup loss can be written as*

$$\mathcal{E}_k^{OTmixup}(f) = \frac{1}{\binom{N}{k}}\sum_{\gamma=1}^{\binom{N}{k}}\mathbb{E}_{\theta,\xi}\left[\frac{1}{k}\sum_{i=1}^k\ell(\tilde{y}_i^\gamma + \tilde{\epsilon}_i^\gamma, f(\tilde{x}_i^\gamma + \tilde{\delta}_i^\gamma))\right].$$

The mean $\bar{x}_i^\gamma$ being shifted toward is exactly the mean of the locally-informed distribution $\mathcal{D}_i$. Moreover, the covariance structure of the perturbations is detailed in the proof (simplified in Section G.1) and is now also derived from the local structure of the distribution, inferred from the optimal transport matchings.

*Proof.* This argument is modelled on a proof of Carratino et al. (2020), so we adopt analogous notation and highlight the differences in our setting and refer the reader to Appendix B.1 of that paper for any omitted details. First, let us use shorthand notation for the interpolated loss function:

$$m_i^{\gamma\xi}(\lambda) = \ell(f(\lambda x_i^\gamma + (1-\lambda)x_{\sigma_{\gamma\xi}(i)}^\xi), \lambda y_i^\gamma + (1-\lambda)y_{\sigma_{\gamma\xi}(i)}^\xi).$$

Then the mixup objective may be written as:

$$\mathcal{E}_k^{mix}(f) = \frac{1}{k\binom{N}{k}^2} \sum_{\gamma,\xi=1}^{\binom{N}{k}} \sum_{i=1}^{k} \mathbb{E}_\lambda m_i^{\gamma\xi}(\lambda).$$

As $\lambda \sim \beta(\alpha,\alpha)$, we may leverage the symmetry of the sampling function and use a parameter $\theta \sim \beta_{[1/2,1]}(\alpha,\alpha)$ to write the objective as:

$$\mathcal{E}_k^{mix}(f) = \frac{1}{\binom{N}{k}} \sum_{\gamma=1}^{\binom{N}{k}} \ell_i, \qquad \text{where } \ell_i = \frac{1}{k\binom{N}{k}} \sum_{\xi=1}^{\binom{N}{k}} \sum_{i=1}^{k} \mathbb{E}_\theta m_i^{\gamma\xi}(\theta)$$

To obtain the form of the theorem in the text, we introduce auxiliary variables $\tilde{x}_i^\gamma, \tilde{y}_i^\gamma$ to represent the mean-reverted training points:

$$\tilde{x}_i^\gamma = \mathbb{E}_{\theta,\xi}\left[\theta x_i^\gamma + (1-\theta)x_{\sigma_{\gamma\xi}(i)}^\xi\right]$$
$$\tilde{y}_i^\gamma = \mathbb{E}_{\theta,\xi}\left[\theta y_i^\gamma + (1-\theta)y_{\sigma_{\gamma\xi}(i)}^\xi\right],$$

and $\tilde{\delta}_i^\gamma, \tilde{\epsilon}_i^\gamma$ to denote the zero mean perturbations about these points:

$$\tilde{\delta}_i^\gamma = \theta x_i^\gamma + (1-\theta)x_{\sigma_{\gamma\xi}(i)}^\xi - \mathbb{E}_{\theta,\xi}\left[\theta x_i^\gamma + (1-\theta)x_{\sigma_{\gamma\xi}(i)}^\xi\right]$$
$$\tilde{\epsilon}_i^\gamma = \theta y_i^\gamma + (1-\theta)y_{\sigma_{\gamma\xi}(i)}^\xi - \mathbb{E}_{\theta,\xi}\left[\theta y_i^\gamma + (1-\theta)y_{\sigma_{\gamma\xi}(i)}^\xi\right].$$

These reduce to the simplified expressions given in the theorem if we recall that $\theta$ and $\xi$ are independent random variables. Note that both the mean-reverted points and the perturbations are informed by the local distribution $\mathcal{D}_i$. $\qquad\square$

### G.1 COVARIANCE STRUCTURE

As in (Carratino et al., 2020), it's possible to come up with some simple expressions for the covariance structure of the local perturbations, so we write out the analogous result below. As the argument is very similar, we omit it.

**Lemma 2.** *Let $\sigma^2$ denote the variance of $\beta_{[1/2,1]}(\alpha,\alpha)$, and $\nu^2 := \sigma^2 + (1-\bar{\theta})^2$. Then the following expressions hold for the covariance of the zero mean perturbations:*

$$\mathbb{E}_{\theta,\xi}\tilde{\delta}_i^\gamma(\tilde{\delta}_i^\gamma)^\top = \frac{\sigma^2(\tilde{x}_i^\gamma - \bar{x}_i^\gamma)(\tilde{x}_i^\gamma - \bar{x}_i^\gamma)^\top + \nu^2\Sigma_{\tilde{x}_i^\gamma\tilde{x}_i^\gamma}}{\bar{\theta}^2}$$

$$\mathbb{E}_{\theta,\xi}\tilde{\epsilon}_i^\gamma(\tilde{\epsilon}_i^\gamma)^\top = \frac{\sigma^2(\tilde{y}_i^\gamma - \bar{y}_i^\gamma)(\tilde{y}_i^\gamma - \bar{y}_i^\gamma)^\top + \nu^2\Sigma_{\tilde{y}_i^\gamma\tilde{y}_i^\gamma}}{\bar{\theta}^2}$$

$$\mathbb{E}_{\theta,\xi}\tilde{\delta}_i^\gamma(\tilde{\epsilon}_i^\gamma)^\top = \frac{\sigma^2(\tilde{x}_i^\gamma - \bar{x}_i^\gamma)(\tilde{y}_i^\gamma - \bar{y}_i^\gamma)^\top + \nu^2\Sigma_{\tilde{x}_i^\gamma\tilde{y}_i^\gamma}}{\bar{\theta}^2},$$

*where $\Sigma_{\tilde{x}_i^\gamma\tilde{x}_i^\gamma}, \Sigma_{\tilde{y}_i^\gamma\tilde{y}_i^\gamma}, \Sigma_{\tilde{x}_i^\gamma\tilde{y}_i^\gamma}$ denote empirical covariance matrices.*

Note again, that the covariances above are locally-informed, rather than globally determined. Lastly, there is also a quadratic expansion performed about the mean-reverted points $\tilde{x}_i^\gamma, \tilde{y}_i^\gamma$ with terms that regularize $f$, but we omit this result as the regularization of Theorem 3 is more intuitive (c.f. Theorem 2 of (Carratino et al., 2020)).

