# OpenReview forum: "$k$-Mixup Regularization for Deep Learning via Optimal Transport"
_ICLR.cc/2022/Conference — ICLR 2022 Submitted_

### Official Review · Reviewer_8N9h · 2021-10-23

**Correctness:** 4
**Technical Novelty And Significance:** 3
**Empirical Novelty And Significance:** 2
**Recommendation:** 3
**Confidence:** 4

**Main Review:**

-Strengths:

1. This paper is generally well-written and well-organized, making it easy to understand.

2. The authors did many experiments in various settings. They used neural networks with different sizes (small MLP to DenseNet) on different datasets (both synthetic and real-world datasets) from different modalities (image and audio). Each experiment is also repeated multiple times.

3. The authors provided very detailed settings, e.g., network structures and hyperparameters, for their experiments. This makes their experiments quite reproducible.

-Concerns:

1. The justification for the proposed k-mixup method might be unclear, and this is my major concern for this paper. I will explain my concern in detail in the following paragraphs:

1.1 The authors claimed that $k$-mixup preserves the local structures of the training data, but this alone is not enough to explain why $k$-mixup could help. Imagine the extreme case where we take $k$ to infinity, and this will essentially be equivalent to having no mixup at all because every data point will mix up with itself. Therefore, there might be deeper reasons why $k$-mixup could work, and this could be related to how the labels are interpolated during mixup. Since $k$-mixup encourages closer points to mix, the mixup between points from different classes will have a sharper label transition compared to the normal mixup. The authors proved in Theorem 2 that $k$-mixup can help the model with smooth interpolation between the clusters, but this model seems somewhat specific and the authors did not talk much about this smoothing effect in this paper.

1.2 Both smaller $\alpha$ and larger $k$ can make the training set become more vicinal, and intuitively it could be unclear which has a better regularization effect. For instance, setting both $k$ and $\alpha$ to be large and setting both to be small should both result in a vicinal dataset. In this case, the two regimes might have a similar local structure for the training data, and it seems confusing why this extra parameter $k$ could help because we can always choose a smaller $\alpha$ to make the training data more local.

1.3 For larger $k$, more training data will be mixed with points within the same class if the data have some cluster structure, as can be seen in Figure 2. Thus, the ratio of the same-class mixup will vary when $k$ changes. This may also influence the regularization effect of k-mixup and it could be possible that changing the ratio of same-class mixup can already improve the performance of mixup.

2. The empirical benefit provided by $k$-mixup might not be significant enough, and this $k$-mixup method requires extra tuning for the hyperparameter $k$. Details are provided below:

2.1 The performance gain by performing $k$-mixup is not consistent for different datasets and network structures. For instance, in the small $\alpha$ regime ($\alpha$=0.1), larger $k$ always does not improve the performance in most of the datasets while requiring more computation and parameter tuning. This could be because smaller $\alpha$ better preserves the local structure of the data and larger $k$ is not needed.

2.2 For the cases where $k$-mixup does improve the performance over the original mixup, the performance gain does not seem to be very large (usually less than 1%), and achieving this performance gain requires much work in tuning the hyperparameters $k$ and $\alpha$. As mentioned in concern 1.2, the regularization effect provided by $k$-mixup is controlled by both $k$ and $\alpha$, and based on the experimental results there seems not to be a consistent scaling law for the best $k$ and $\alpha$. Specifically, from the best-performance model for original mixup($k$=1), achieving best performance sometimes require us to increase both $k$ and $\alpha$, but sometimes will require increasing $k$ and decreasing (or keep) $\alpha$ instead, depending on the task. This probably means tuning $k$-mixup requires a grid search over $k$ and $\alpha$, which introduces extra computation cost.

2.3 It would be better if the authors could compare $k$-mixup to other mixup variants. This paper only provides one experiment comparing $k$-mixup to manifold mixup on one task, so it might not be convincing enough that $k$-mixup could perform better than other variants of mixup.

2.4 The authors claimed in their paper that the extra computation cost caused by $k$-mixup is small, and it could be better if they could provide numerical evidence for this, e.g., compare the wallclock time of training the same model using regular mixup to that of $k$-mixup where $k$=32.

3. The assumptions and conclusions of the theoretical claims are somewhat unrealistic.

3.1 The theoretical claims often require k to be "large enough", which actually needs $k$ to be larger than some exponential function of dimension $d$. This cannot be true in practice. For Proposition 1, we need $k$ to be larger than both $\Omega(1/\delta)^d$ and $(1/R_S)^d$. For Theorem 2, $k$ needs to ensure both $A_\delta$ and $B_\delta$ contain enough points, which could also require $k$ to be exponentially large. This is somewhat unrealistic because the inputs for the usual tasks are usually of high dimension. It would also be better if the authors could explicitly state the requirement of $k$ in their statements of theorems.

3.2 The assumptions needed for the theoretical results might lack justifications. For section 3.2, the authors assumed that the input data to be $(m, \Delta)$-clusterable and the distance between any pair of covering balls is at least $2\Delta$. This assumption intuitively would result in a very large $m$ for real data and largely weaken the conclusions. Besides, it might be better if the authors could explicitly state this assumption outside the Lemma since Theorem 1 and 2 also need that assumption.

3.3 Section 4 might seem a bit confusing. The expressions for the loss and regularizers are roughly the same as the previous paper (Zhang et al., 2020) except that the expectation in the regularization terms is taken over the "locally-informed distribution", but it seems unclear why this is better than the original distribution.

-Minor Comments:

1. Figure 1 seems a bit confusing and might need more interpretation. The authors claimed that it shows $k$-mixup can better keep the manifold support structure, but it seems that 4-mixup produces a more blurry function on Swiss Roll than 1-mixup, which seems confusing. Besides, it might be better if the authors could provide more detailed explanations about these datasets, e.g., how they are generated and labeled, to make Figure 1 more clear.

2. Some of the intuitions from the paper might become weaker for higher-dimensional data. For example, in the regime when the number of data is not much larger than the dimension, the data points will become more separated in the sense that linear interpolation between any two points might not be close to any other points. This regime may be very different from the synthetic datasets visualized in Figures 1 and 2. Besides, since the theoretical results in this paper often require $k$ to be exponentially large in $d$, this could also be part of the reason why $k$-mixup doesn't work so well in high dimensions. The authors discussed this in section 6, and it might be better if the authors could comment more.

3. For the Toy datasets, especially One Ring and Swiss Roll, normal mixup seems to work better for smaller $\alpha$, so perhaps the authors could try to use even smaller $\alpha$ and see whether that can give even better results.

4. For the performance on Google Speech Commands, the authors claimed that the difference between the best $k$-mixup and the best normal mixup is 0.6508%. How is this number computed?

-Typos: There is a duplicate sentence near the end of Section 2: "The localized nature of the matchings makes it more likely that the averaged labels will smoothly interpolate over the decision boundaries." appears twice.

---------------------Update--------------------------

I have read all the other reviews, the authors' responses to all the reviewers, and the revisions the authors made in the updated draft, and I have decided to keep my score unchanged, i.e., I tend to recommend rejection. I would like to thank the authors for their very detailed response which addressed some of my concerns (e.g., ratio of same-class mixup, wallclock time comparison), but my major concerns (unclear justification, marginal performance gain) about this paper still remain. Detailed reasons why I keep my score are listed below:

- The justification for $k$-mixup seems somewhat unclear, and this is a common concern for most of the reviewers (P7xo, iZL9, K2Mi, 8N9h). The authors claimed that this method can better preserve cluster and manifold structures and provide some smoothing effect between clusters. However, both claims might be too abstract. The authors provided theoretical explanations in Section 3, but the implications for these theorems seem unclear, e.g., as also mentioned by reviewer P7xo and K2Mi, they cannot explain why increasing k doesn't necessarily always improve the performance.

- The performance improvements on real datasets seem marginal, and this is also a common concern for most reviewers (iZL9, EGaJ, 8N9h). Besides, getting this performance gain requires tuning an additional parameter $k$, and there is no simple indicator (e.g., one cannot use average squared distance to directly predict the performance gain) for this gain, so one really needs to train more models with different $k$'s and $\alpha$'s and do cross-validation, which requires much more computation power.

**Summary Of The Paper:**

This paper proposed $k$-mixup, which is a generalization of mixup, to better regularize neural networks during training. Specifically, instead of mixing totally random pairs of training data, the authors draw $k$ pairs of data each time and mix them in a way that minimizes the Wasserstein distance between them. The authors theoretically proved that their method can better preserve the local structure of the data. They also did experiments for various networks and datasets and showed that $k$-mixup can improve the generalization performance and robustness of the models compared to the original mixup.

**Summary Of The Review:**

I tend to vote for rejecting this paper. Despite being clearly written, the justification for their proposed method is somewhat unclear, and the performance gain seems not significant and consistent enough. Therefore, I think more work needs to be done to provide enough justifications for $k$-mixup (perhaps from the theoretical side) and to further improve the empirical performances.

---

> ### Author Response · Authors · 2021-11-16
> **Thank you, and responses (1)**
>
> Thank you for the comments and close reading of the work. We especially appreciate that you found the paper to be well written and organized, with detailed and reproducible experiments. Here are some responses to your questions and remarks.
>
> > 1.1 The authors claimed that $k$-mixup preserves the local structures of the training data, but this alone is not enough to explain why $k$-mixup could help. Imagine the extreme case where we take $k$ to infinity, and this will essentially be equivalent to having no mixup at all because every data point will mix up with itself. Therefore, there might be deeper reasons why $k$-mixup could work, and this could be related to how the labels are interpolated during mixup. Since $k$-mixup encourages closer points to mix, the mixup between points from different classes will have a sharper label transition compared to the normal mixup. The authors proved in Theorem 2 that $k$-mixup can help the model with smooth interpolation between the clusters, but this model seems somewhat specific and the authors did not talk much about this smoothing effect in this paper.
>
> We will make clearer our discussion of this smoothing effect and $k$-mixup encouraging closer points to mix in the paper, as these are indeed the primary reasons we advocate for $k$-mixup throughout the paper. These claims are exactly what we mean by “preserves the local structures of the training data.” Where different classes have points nearby, one expects a sharper label transition to interpolate well between the classes.
>
> > Imagine the extreme case where we take $k$ to infinity, and this will essentially be equivalent to having no mixup at all because every data point will mix up with itself.
>
> This is not the case due to optimal transport constraints. The $k$-batches are sampled without replacement, so there should be no shared elements. Also, see Theorems 1 and 2 and subsequent discussion showing an increasing absolute number of cross-cluster matches as $k$ increases. Lastly, in all our experiments $k \ll N$, so couplings will be locally-informed, but will still be non-trivial.
>
> > Both smaller $\alpha$ and larger $k$ can make the training set become more vicinal, and intuitively it could be unclear which has a better regularization effect. For instance, setting both and to be large and setting both to be small should both result in a vicinal dataset. In this case, the two regimes might have a similar local structure for the training data, and it seems confusing why this extra parameter could help because we can always choose a smaller alpha to make the training data more local.
>
> The purpose of larger $k$ is not at all to make the training distribution more local (otherwise small $\alpha$ would always be better), but instead to make the training distribution locally informed. Specifically, the goal is to respect cluster and manifold structure, so that many well-scattered (not local!) new points are introduced in the voids between clusters and manifolds, which regularize the neural network to have smooth transitions between clusters and classes. Most importantly, these well-scattered points in the voids should have interpolated labels that reflect the labels of the clusters and manifolds closest to them, something that $k$-mixup provides but 1-mixup cannot (see Figure 1 etc.).
>
> > 1.3 For larger $k$, more training data will be mixed with points within the same class if the data have some cluster structure, as can be seen in Figure 2. Thus, the ratio of the same-class mixup will vary when $k$ changes. This may also influence the regularization effect of $k$-mixup and it could be possible that changing the ratio of same-class mixup can already improve the performance of mixup.
>
> The prospect of fixing the ratio of cross-class and in-class mixup is certainly an idea that we considered, but did not prove beneficial in initial experiments. It is perhaps an avenue for future exploration though. We can mention this in future work.
>
> > 2.1 The performance gain by performing $k$-mixup is not consistent for different datasets and network structures. For instance, in the small $\alpha$ regime ($=0.1$), larger $k$ always does not improve the performance in most of the datasets while requiring more computation and parameter tuning. This could be because smaller better preserves the local structure of the data and larger $k$ is not needed.
>
> We point out that $\alpha$ is not an independent parameter, it is a tuning parameter just like $k$ is. Hence it is not appropriate to discuss the “small $\alpha$ regime,” instead one should compare the best $\alpha$ for $k=1$ vs. the best $\alpha$ vs. $k > 1$.

---

> ### Author Response · Authors · 2021-11-16
> **Responses (2)**
>
> > For the cases where $k$-mixup does improve the performance over the original mixup, the performance gain does not seem to be very large (usually less than 1%), and achieving this performance gain requires much work in tuning the hyperparameters $k$ and $\alpha$. As mentioned in concern 1.2, the regularization effect provided by $k$-mixup is controlled by both $k$ and $\alpha$, and based on the experimental results there seems not to be a consistent scaling law for the best $k$ and $\alpha$. Specifically, from the best-performance model for original mixup ($=1$), achieving best performance sometimes require us to increase both $k$ and $\alpha$, but sometimes will require increasing $k$ and decreasing (or keep) $\alpha$ instead, depending on the task. This probably means tuning $k$-mixup requires a grid search over and, which introduces extra computation cost.
>
> Performance gain: We point out that mixup variants typically do not see spectacular gains, so it is not surprising that our approach sees moderately smaller gains on top of that. In spite of this and the additional $\alpha$ parameter that should be tuned, mixup techniques are widely used. This may be due to the fact that in many applications, achieving the best and most robust performance possible is much more important than saving computation (e.g. small to medium-sized models deployed in high-volume or high-impact settings).
>
> Choosing $k$: While there is not a consistent pattern as to which $k$ achieves the best performance, note that even $k=2$ always outperforms $k=1$ except in two instances in the experiments. Hence in most cases it should not be strictly necessary to tune $k$ if the additional computation is prohibitive. When additional computation is available, however, $k$ can be chosen with cross-validation just as alpha must be with standard mixup. We found it sufficient to try values of $k$ that are powers of 2, potentially making cross-validation simpler.
>
> > 2.3 It would be better if the authors could compare $k$-mixup to other mixup variants. This paper only provides one experiment comparing $k$-mixup to manifold mixup on one task, so it might not be convincing enough that $k$-mixup could perform better than other variants of mixup.
>
> We also have considered comparison to AdaMix, but were unable to find nicely structured publicly available code. Other mixup variants, like CutMix are specific to image domains. Lastly, all of these are far more complicated and computationally expensive than our methods.
> The authors claimed in their paper that the extra computation cost caused by $k$-mixup is small, and it could be better if they could provide numerical evidence for this.
>
> When using $k=32$ on CIFAR, the cost of $k$-mixup adds 8 seconds per epoch. The vast majority of this time is spent in constructing the cost matrix for the OT problem (done on a CPU) and could be optimized with a parallelized GPU computation. We will add a sentence making this small cost more specific.
>
> > 3.1 The theoretical claims often require $k$ to be "large enough", which actually needs $k$ to be larger than some exponential function of dimension. This cannot be true in practice. For Proposition 1, we need k to be larger than both $\Omega(1/\delta)^d$ and $(1/R_S)^d$. For Theorem 2, needs to ensure both and contain enough points, which could also require to be exponentially large. This is somewhat unrealistic because the inputs for the usual tasks are usually of high dimension. It would also be better if the authors could explicitly state the requirement of $k$ in their statements of theorems.
>
> We only require large enough $k$ for two results, Proposition 1 and Theorem 2. Both are intended to show a limiting result, in the sense of asymptotic consistency akin to many results in machine learning theory. Neither result makes any implication that large $k$ is needed in practice, their purpose is simply to establish that $k \to \infty$ makes sense from a regularization perspective. Additionally, there is no need in Theorem 2 for $k$ to be exponentially large, as it is not necessary to drive $\epsilon$ to zero and in any case the sets $A_\epsilon$ are not $\epsilon$-spheres, and should have volume scaling approximately linearly with $\epsilon$ regardless of dimension.

---

> ### Author Response · Authors · 2021-11-16
> **Responses (3)**
>
> > 3.2 The assumptions needed for the theoretical results might lack justifications. For section 3.2, the authors assumed that the input data to be $(m,\Delta)$ - clusterable and the distance between any pair of covering balls is at least $2\Delta$. This assumption intuitively would result in a very large $m$ for real data and largely weaken the conclusions. Besides, it might be better if the authors could explicitly state this assumption outside the Lemma since Theorem 1 and 2 also need that assumption.
>
> We will clarify Theorem 1 by adding the assumption from Lemma 1. We also point out that for Theorem 1, the ball construction is introduced as a worst-case setting. If the clusters are closer together than assumed, more cross-cluster matches should be expected, not fewer! Hence for real data the conclusions will be strengthened, not weakened. We will clarify this in the revision. Theorem 2 is separate from Lemma 1 and Theorem 1, in particular Theorem 2 does not use $(m,\Delta)$ clusterability at all.
>
> > 3.3 Section 4 might seem a bit confusing. The expressions for the loss and regularizers are roughly the same as the previous paper (Zhang et al., 2020) except that the expectation in the regularization terms is taken over the "locally-informed distribution", but it seems unclear why this is better than the original distribution.
>
> We believe that while by necessity similar, these results are important to include for completeness. The “locally-informed distribution” is a direct reference to discussions throughout the paper as to why the locally-informed $k$-mixup is advantageous. We will clarify this point in the revision.
>
> > Figure 1 seems a bit confusing and might need more interpretation. The authors claimed that it shows $k$-mixup can better keep the manifold support structure, but it seems that 4-mixup produces a more blurry function on Swiss Roll than 1-mixup, which seems confusing. Besides, it might be better if the authors could provide more detailed explanations about these datasets, e.g., how they are generated and labeled, to make Figure 1 more clear.
>
> First, it should be noted that there is randomness in the sampling of $k$-batches, and thus the couplings used to generate the vicinal distributions. This leads to some degree of randomness in the training of the models, so the poorer label smoothing may be a result of that here. Furthermore, the Swiss Roll consists of two heavily entangled classes, so it is possible that $k=4$ is just not high enough to respect the manifold and cluster structure here. One can see that $k=32$ clearly improves upon this greatly. We will run some additional experiments to investigate this.
>
> > Some of the intuitions from the paper might become weaker for higher-dimensional data. For example, in the regime when the number of data is not much larger than the dimension, the data points will become more separated in the sense that linear interpolation between any two points might not be close to any other points. This regime may be very different from the synthetic datasets visualized in Figures 1 and 2. Besides, since the theoretical results in this paper often require $k$ to be exponentially large in $d$, this could also be part of the reason why $k$-mixup doesn't work so well in high dimensions. The authors discussed this in section 6, and it might be better if the authors could comment more.
>
> We agree that large dimension may result in smaller improvement of $k$-mixup, in fact this is generally what we see in the experiments. That said, the improvement in extremely high dimensional settings such as CIFAR-10 (1024 dimensions) is still statistically significant as the experiments demonstrate. The results showing that larger $k$ does indeed succeed in finding closer matches (the average squared distance tables) are indicative that at least some of the lower-dimensional intuition survives. Additionally, while the ambient dimensions may be extremely large, it is well known that high dimensional data often lives on low-dimensional manifolds, which should alleviate some of these high dimension concerns.
>
> > For the Toy datasets, especially One Ring and Swiss Roll, normal mixup seems to work better for smaller $\alpha$, so perhaps the authors could try to use even smaller $\alpha$ and see whether that can give even better results.
>
> We will include the no mixup performance in a revision, and evaluate whether smaller $\alpha$ improves the results.
>
> > For the performance on Google Speech Commands, the authors claimed that the difference between the best $k$-mixup and the best normal mixup is 0.6508%. How is this number computed?
>
> This seems to be a typo, we’ve corrected it.

---

> ### Author Response · Authors · 2021-11-16
> **Responses (4)**
>
>
> > -Typos: There is a duplicate sentence near the end of Section 2: "The localized nature of the matchings makes it more likely that the averaged labels will smoothly interpolate over the decision boundaries." appears twice.
>
> Thanks for pointing this out.
>
> We hope these answers are convincing, and will be glad to clarify further if needed. If they bolster your estimation of the work, please consider modifying your score to reflect this.

---

> > ### Comment · Reviewer_8N9h · 2021-11-20
> > **Update my review after author response**
> >
> > Thank the authors for the very detailed response! I have updated my review (See the contents in the "Update" section at the bottom of "Main Review"). Although some of my concerns are addressed by the responses, my major concerns remain, so I kept my score unchanged. Detailed reasons for this are listed in my updated review.

---

### Official Review · Reviewer_K2Mi · 2021-11-02

**Correctness:** 4
**Technical Novelty And Significance:** 3
**Empirical Novelty And Significance:** 2
**Recommendation:** 6
**Confidence:** 2

**Details Of Ethics Concerns:**

None.

**Main Review:**

Pros:
- This work is well-motivated and well-written.
- Theoretical study supports the claim of the paper (as k increases, the vicinal samples better reflect the local structure of the dataset).
- Experimental protocol seems sound and varied.

Cons:
- There are two hyper-parameters, k and alpha, whose choice is not completely clear to me (see below).

Questions and remarks:
- It could be useful to the reader to elaborate on the advantage of the Hungarian algorithm over Sinkhorn w.r.t k.
- Do you really need to compare the cost of the regularization to the cost of computing gradients? Isn't k-mixup regularization a pre-processing step with "fixed cost"?
- Do you have an idea why increasing k does not always improve your results? This seems to be opposed to the intuition that higher k better reflects the structure of the dataset. More generally, it could be great to have a discussion on how to choose k and alpha for your method.
- Could this technique be useful in the context of transfer learning (I am not asking for more experiments here)?

**Summary Of The Paper:**

This work proposes an improvement on the Mixup regularization for training deep neural networks. Instead of performing weighted averages of randomly chosen pairs of samples, an optimal transport map between two k-batches is computed. Then, "new" samples are constructed by interpolating between *coupled* pairs of samples (according to the optimal transport plan). This enables to better reflect the local structure of the dataset. A theoretical study supports this intuition, and extensive experiments are conducted.

**Summary Of The Review:**

This work seems well-motivated, clearly explained with globally convincing experiments. My only concern is on how to choose the hyper-parameters k and alpha, which do not seem to always have the intended effect. Overall, I tend to recommend acceptance.

---

> ### Author Response · Authors · 2021-11-16
> **Response to Reviewer K2Mi**
>
> Thank you for the comments and close reading of the work. We appreciate the positive evaluation, and the view that our approach is well-motivated and backed up with convincing experiments. Here are some responses to your questions and remarks.
>
> **It could be useful to the reader to elaborate on the advantage of the Hungarian algorithm over Sinkhorn w.r.t k.**
>
> For large k, Sinkhorn is O(k log k) amortized cost versus O(k^2) for Hungarian. However, for small k the cost of the Hungarian algorithm is trivial, at 0.69 seconds per CIFAR epoch when k=32. We will add this.
>
> **Do you really need to compare the cost of the regularization to the cost of computing gradients? Isn't k-mixup regularization a pre-processing step with "fixed cost"?**
>
> K-mixup can indeed be implemented as a fixed-cost pre-processing step, but we found (as did the original mixup paper) advantages to performing the mixup operation anew every epoch “in-line”. The comparison to gradient computation is mostly to point out how the small computational cost of k-mixup has no bearing on the overall high computational cost of training neural network models.
>
> **Do you have an idea why increasing k does not always improve your results? This seems to be opposed to the intuition that higher k better reflects the structure of the dataset. More generally, it could be great to have a discussion on how to choose k and alpha for your method.**
>
> Choosing k: While there is not a consistent pattern as to which k achieves the best performance, note that even k=2 always outperforms k=1 except in two instances in the experiments. Hence in most cases it should not be strictly necessary to tune k if the additional computation is prohibitive. When additional computation is available, however, k can be chosen with cross-validation just as alpha must be with standard mixup. We found it sufficient to try values of k that are powers of 2, potentially making cross-validation simpler.
>
> Why increasing k does not always improve results: Based on our theory, increasing k should make the transitions between clusters smoother. While this should intuitively tend to improve generalization performance, whether this actually happens is dependent on the structure of the test data distribution, particularly as mixup improvements generally involve 1% change or less. As a result, there is no reason to believe that finding better smoothed loss (or increasing k) will have a monotonic effect on generalization performance for any given problem. Instead, what we tend to see is that large k often does well on classification tasks (due to the high degree of smoothness), but moderate k can outperform it.
>
> **Could this technique be useful in the context of transfer learning (I am not asking for more experiments here)?**
>
> This is an intriguing connection! Indeed some sort of optimal transport mapping from the original data to a target dataset could perhaps allow for e.g. labels generated by the source model on the source data to the target data points.
>
> We hope these answers are convincing, and will be glad to clarify further if needed.

---

> > ### Comment · Reviewer_K2Mi · 2021-11-22
> > **Thank you but I still have concerns**
> >
> > I have read the other reviews as well as author response to my review and other author response. First, I want to thank the authors for the additional details. Then, I am still concerned by the fact that there is no consistent way of setting k and alpha, except cross-validation, which may indeed be too computationally expensive compared to the marginal improvements, as observed for example by reviewer 8N9h. Hence, I will decrease my score although I still tend to recommend acceptance.

---

### Official Review · Reviewer_EGaJ · 2021-11-05

**Correctness:** 3
**Technical Novelty And Significance:** 2
**Empirical Novelty And Significance:** 2
**Recommendation:** 6
**Confidence:** 3

**Main Review:**

This paper shows some nice improvements on low-dimensional toy datasets from using optimal transport to select examples from two different batches to interpolate.  The improvements on larger datasets are more marginal, but are still significant when the mixing rate alpha is large.  This is decent work and it may have some impact, but the small improvements may make the impact fairly limited.  Additionally what it achieves does overlap some with Manifold Mixup, although the paper does explain that Manifold Mixup has other drawbacks.

Other comments:
  -First paragraph of intro is really boilerplate that could be removed.

  -Figure 1 is a bit weaker of a result than could be possible - since the solution inside the spiral is still somewhat but only partially blurred after using k-mixup.  Nonetheless it is better than the baseline.

  -Figure 2 is pretty convincing that more points are being interpolated onto the same manifold when using a larger K.

  -How is the algorithm different from mixing with nearest neighbors from a limited pool of examples?  Perhaps it's the requirement of an optimal transport (so that the same point can't be picked twice as a neighbor)?

  -The insight in Theorem 1 is nice, especially that cross-cluster mixes will still be selected more as K grows, but as a decreasing fraction of K.

  -The classification results (Figure 8) are a bit discouraging, and somewhat contradict the introduction which claims that the k-mixup technique doesn't hurt results, when several results are 0.1-0.3 basis points below the baseline.  Still, where improvements occur they are often of larger magnitude than the deteriorations.

  -The improvement with large alpha is impressive.

  -FGSM is a very weak attack, so the improvements in Figure 12 are of questionable significance, although it is nice to see slightly better robustness.



**Summary Of The Paper:**

This paper proposes to select pairs of points to mix between two batches of K examples by using the hungarian algorithm to find an L2 optimal matching.  This substantially improves performance on non-linear low-dimensional classification tasks where mixup underfits and it also improves results on moderate-scale classification tasks (like CIFAR-100) and especially helps when the mixing rate alpha is large.  There is also theory analyzing how often mixing will interpolate between different clusters when using k-mixup.

Detailed Notes from reading the paper:

  -Modification of mixup where batches of k points are perturbed in the direction of k other points using interpolation under a Wasserstein metric.

  -Proof with experiments and theory showing this k-mixup preserves cluster and manifold structure.

  -K-mixup improves results (or keeps same) and improves adversarial robustness.

  -Increasing K makes interpolated points more likely to be on data manifold.

  -Use Hungarian algorithm to find L2 optimal permutation.



**Summary Of The Review:**

This paper achieves small improvements on large datasets and significant improvements on either low dimensional data or where the mixing rate alpha is very large.  The improvements are small but the idea is simple and logical, so I weakly recommend acceptance.

---

> ### Author Response · Authors · 2021-11-16
> **Response to Reviewer EGaJ**
>
> Thank you for the comments and close reading of the work. We appreciate the positive evaluation and appreciation of the logic of the approach. Here are some responses to your questions and remarks.
>
> **Figure 1 is a bit weaker of a result than could be possible - since the solution inside the spiral is still somewhat but only partially blurred after using k-mixup. Nonetheless it is better than the baseline.**
>
> First, it should be noted that there is randomness in the sampling of k-batches, and thus the couplings used to generate the vicinal distributions. This leads to some degree of randomness in the training of the models, so the poorer label smoothing may be a result of that here. Furthermore, the Swiss Roll consists of two heavily entangled classes, so it is possible that k=4 is just not high enough to respect the manifold and cluster structure here. One can see that k=32 clearly improves upon this greatly. We will run some additional experiments to investigate this.
>
> **How is the algorithm different from mixing with nearest neighbors from a limited pool of examples? Perhaps it's the requirement of an optimal transport (so that the same point can't be picked twice as a neighbor)?**
>
> Yes, this is the primary difference. A significant advantage of the requirement of optimal transport is that cross-cluster matches become not only possible but highly likely, which are critical for learning smooth transitions between clusters. A nearest-neighbor approach within a k-sample would be very unlikely to have *cross-cluster matches* once k becomes larger than the number of clusters.
>
> **The classification results (Figure 8) are a bit discouraging, and somewhat contradict the introduction which claims that the k-mixup technique doesn't hurt results, when several results are 0.1-0.3 basis points below the baseline. Still, where improvements occur they are often of larger magnitude than the deteriorations.**
>
> We point out that our claim of “k-mixup doesn’t hurt results” refers to the fact that tuned k-mixup (i.e. tuned alpha and k > 1) always matches or outperforms tuned 1-mixup. We will add this clarification in the paper.
>
> **FGSM is a very weak attack, so the improvements in Figure 12 are of questionable significance, although it is nice to see slightly better robustness.**
>
> In this experiment, we went with a weak gradient-based attack following the approach of the original Mixup paper. The goal of the experiment is to show additional robustness to specifically the gradient-type attack, since this is the exact type of attack that attempts to exploit a lack of a smooth transition between classes (which (k-)mixup provides). In the revision, we will make this point clearer.
>
> We hope these answers are convincing, and will be glad to clarify further if needed. If they bolster your estimation of the work, please consider adjusting your score to reflect this.

---

### Official Review · Reviewer_iZL9 · 2021-11-05

**Correctness:** 2
**Technical Novelty And Significance:** 3
**Empirical Novelty And Significance:** 2
**Recommendation:** 5
**Confidence:** 4

**Main Review:**

Major comments:

-- Proposition 1 and Theorem 1's implications are unclear. Does this mean that k-mixup with large k is closer to "on-manifold mixup" such as SMOTE?  For instance, see "[Chawla et al. (2002)] that proposed to augment the rare class in an imbalanced dataset by interpolating the nearest neighbors and [DeVries & Taylor (2017)] that showed that interpolation and extrapolation the nearest neighbors of the same class in feature space can improve generalization". (The description of these papers are excerpted from the original mixup paper -- the authors may want to refer to Sec. 4 of the original paper.)

-- "The localized nature of the matchings makes it more likely that the averaged labels will smoothly interpolate over the decision boundaries. A consequence is that k-mixup is robust to higher values of α, since it is no longer necessary to keep λ close to 0 or 1 to avoid erroneous labels." => For high-dimensional data, even with higher values of alpha, it is unclear if erroneous labels (aka manifold intrusion) will occur with high probability. This questions the practical gain of the proposed method on high-dimensional data.

-- The proposed idea seems relevant to ["GAN-mixup: Augmenting Across Data Manifolds for Improved Robustness", Sohn et al.].  The authors may want to clarify the difference between the proposed approach and the one introduced in this prior work.

-- Performance comparisons with baseline algorithms are missing.
--- Guo et al.'s AdaMixup
-- ["On Adversarial Mixup Resynthesis", Beckham et al.]

-- Some performance gains (compared to k = 1) on real datasets look too marginal; See Figure 10, Figure 11

-- By looking at Figure 5 and Figure 6, the performance seems to be highly correlated with the average squared distance of vicinal distribution from training set, regardless of the choice of k.  For instance, in Figure 6, (k=1, alpha=0.5) and (k=8, alpha=1) have almost the same distance as well as the test accuracy.

This makes it unclear whether or not the performance gain actually comes from the benefits of k-mixup. Instead, this might be an artifact of decreasing distance intervals as k increases. For instance, in Figure 5, even though the same values of alpha's are tested, k=1's average squared distance ranges from 0.068 to 1.732, while k=32's those ranges from 0.029 to 0.586. This allows the latter to try out more reasonable values of the squared distances.

To show that this is not the case, the authors may want to rerun the experiments while adaptively setting the range of alpha values for a different choice of k such that the same (or similar) alpha values can be tested.

-- The confidence on test performance seems surprisingly too low to me. What are the random factors across different Monte Carlo runs?  Random initializations and random shuffling usually alone usually give a much higher confidence interval than the reported values such as 0.02 or 0.05.
(See Table 1 in this for instance -- https://arxiv.org/abs/2109.08203)

-- Adversarial robustness: robust accuracy against FGSM (or any simple gradient-based attack) can be highly misleading. Please use AutoAttack by Croce and Hein instead to see whether there exists an actual robustness gain.


Minor comments:

-- "Averaging weights are typically drawn from a beta distribution β(α, α), with parameter α ≪ 1 such that the generated training set is vicinal" -> Not true.  See Table 1 and Table 2 in the original mixup paper for the choice of large alpha values.  Also, the original paper says "For example, in CIFAR-10 classification we can get very low training error on real data even when α → ∞ (i.e., training only on averages of pairs of real examples)".

**Summary Of The Paper:**

The paper proposes an optimal transport-based mixup algorithm, theoretically analyzes the algorithm, and empirically evaluates its performance.

**Summary Of The Review:**

The theoretical claims look solid, but their implications are unclear.  The experimental settings and results could be improved.

---

> ### Author Response · Authors · 2021-11-16
> **Thank you, and responses, Part 1**
>
> Thank you for the comments and close reading of the work. Here are some responses to your questions and remarks.
>
> > Proposition 1 and Theorem 1's implications are unclear. Does this mean that k-mixup with large k is closer to "on-manifold mixup" such as SMOTE? For instance, see "[Chawla et al. (2002)] that proposed to augment the rare class in an imbalanced dataset by interpolating the nearest neighbors and [DeVries & Taylor (2017)] that showed that interpolation and extrapolation the nearest neighbors of the same class in feature space can improve generalization".
>
> When considering only same-class interpolation or interpolation with nearest neighbors (effectively the same thing usually), in contrast to k-mixup, there is little chance of cross-class connections that would lead to label smoothing and interpolation between different classes.
>
> > "The localized nature of the matchings makes it more likely that the averaged labels will smoothly interpolate over the decision boundaries. A consequence is that k-mixup is robust to higher values of α, since it is no longer necessary to keep λ close to 0 or 1 to avoid erroneous labels." => For high-dimensional data, even with higher values of k, it is unclear if erroneous labels (aka manifold intrusion) will occur with high probability. This questions the practical gain of the proposed method on high-dimensional data.
>
> Our theoretical results establish that these manifold-intrusion labels become decreasingly likely for any dimension. Hence our k-mixup approach should be expected to have fewer such erroneous labels than 1-mixup, but of course these are not guaranteed to be zero.
>
> > The proposed idea seems relevant to ["GAN-mixup: Augmenting Across Data Manifolds for Improved Robustness", Sohn et al.]. The authors may want to clarify the difference between the proposed approach and the one introduced in this prior work.
>
> The idea has similar goals, and we offer a much simpler method for achieving these cross-class augmentations. There is no training or use of a GAN needed in our method. We’ve added it as a citation.
>
> > Performance comparisons with baseline algorithms are missing, e.g., Guo et al.'s AdaMixup -- ["On Adversarial Mixup Resynthesis", Beckham et al.]
>
> We also have considered comparison to AdaMix, but were unable to find nicely structured publicly available code. Other mixup variants, like CutMix are specific to image domains. Lastly, all of these are far more complicated and computationally expensive than our methods.
>
> > Some performance gains (compared to k = 1) on real datasets look too marginal; See Figure 10, Figure 11
>
> Recall that Figure 11 is not our proposed method, it is an experiment showing that both manifold mixup and the combination of manifold mixup with our proposed method are inferior to our proposed k-mixup. In Figure 10, the performance gain is 0.4%, well above the margin for statistical significance.
>
> (continued in next remark)

---

> > ### Author Response · Authors · 2021-11-16
> > **Thank you, and responses, Part 2**
> >
> > > By looking at Figure 5 and Figure 6, the performance seems to be highly correlated with the average squared distance of vicinal distribution from training set, regardless of the choice of k. For instance, in Figure 6, (k=1, alpha=0.5) and (k=8, alpha=1) have almost the same distance as well as the test accuracy. This makes it unclear whether or not the performance gain actually comes from the benefits of k-mixup. Instead, this might be an artifact of decreasing distance intervals as k increases. For instance, in Figure 5, even though the same values of alpha's are tested, k=1's average squared distance ranges from 0.068 to 1.732, while k=32's those ranges from 0.029 to 0.586. This allows the latter to try out more reasonable values of the squared distances. To show that this is not the case, the authors may want to rerun the experiments while adaptively setting the range of alpha values for a different choice of k such that the same (or similar) alpha values can be tested.
> >
> > While there is some correlation between average squared vicinal distance and performance for MNIST and CIFAR-10, note that for both types of mixup, the best generalization performance is at neither extreme value of the “average squared vicinal distance”. Hence it is simple to read off the plot and confirm that the performance improvement cannot be explained by the average squared vicinal distance. In particular, for Figure 5, the best performance is achieved at k=32, alpha=100, for which the average squared vicinal distance is 0.586. Looking at the average squared vicinal distances for k=1, we find that this 0.586 value lies between k=1, alpha =.5, and k=1, alpha=1. The classification performance for these values are 99.03 and 98.98, which are inferior to the k=32 performance of 99.18. Similarly, for Figure 6, the best performance is achieved at k=8, alpha=1, with an average squared vicinal distance of 199.5. Again, finding the alpha for which k=1 achieves this average squared vicinal distance, we bookend it with k=1, alpha=0.5 and k=1, alpha=1, which have performances 95.645 and 95.63, inferior to the k=8, alpha=1 performance of 95.815. In fact, by doing a sweep over alpha and choosing the best for each value of k, we have removed the possibility that any of the performance gain of k-mixup over 1-mixup can be explained by the average squared vicinal distance. As this is one of the main reasons to include a table of the average squared distance, we will add a discussion of this to the text.
> >
> > > The confidence on test performance seems surprisingly too low to me. What are the random factors across different Monte Carlo runs? Random initializations and random shuffling usually alone usually give a much higher confidence interval than the reported values such as 0.02 or 0.05. (See Table 1 in this for instance -- https://arxiv.org/abs/2109.08203)
> >
> > Note that the confidence bars are not the single-run confidence, but the confidence after significant Monte Carlo averaging. In other words, these are the confidence intervals on the reported average numbers themselves and should indeed be small. These confidence intervals can then be used to evaluate whether the reported average performance of one method is significantly greater than the reported average for another method or not.
> >
> > > Adversarial robustness: robust accuracy against FGSM (or any simple gradient-based attack) can be highly misleading. Please use AutoAttack by Croce and Hein instead to see whether there exists an actual robustness gain.
> >
> > In this experiment, we follow the approach of the original Mixup paper. The goal is to show additional robustness to specifically the gradient-type attack, since this is the exact type of attack that attempts to exploit a lack of a smooth transition between classes (which k-mixup provides). In the revision, we will make this point clearer.
> >
> > > "Averaging weights are typically drawn from a beta distribution β(α, α), with parameter α ≪ 1 such that the generated training set is vicinal" -> Not true. See Table 1 and Table 2 in the original mixup paper for the choice of large alpha values. Also, the original paper says "For example, in CIFAR-10 classification we can get very low training error on real data even when α → ∞ (i.e., training only on averages of pairs of real examples)".
> >
> > We will correct our description here - the larger point was that alpha << 1 makes the generated training set vicinal.
> >
> > We hope these answers are convincing, and will be glad to clarify further if needed. If they bolster your estimation of the work, please consider modifying your score to reflect this.

---

### Official Review · Reviewer_P7xo · 2021-11-05

**Correctness:** 3
**Technical Novelty And Significance:** 2
**Empirical Novelty And Significance:** 3
**Recommendation:** 3
**Confidence:** 4

**Main Review:**

Description: The paper explains well the method and it can be reproduced easily in my opinion. However, I find it hard to fully understand how they build the entire minibatches of data they use (see questions below). There are some details I am not sure to understand for now and I feel the paper lacks a bit of clarity. May be adding some concrete examples might help readers to get the picture of the creation of the full minibatch used in training or may be adding the pseudo code of their algorithm. Regarding theoretical results, the paper discusses their results properly, however I have some concerns about the validity of the results due to the fact that authors use in practice minibatch optimal transport instead of exact OT (see questions and remarks below). Thus, I am not convinced of the pertinence of some claims.


Evaluation: The evaluation of the methods seems complete experimentally. The method has been used on classification problems on a different datasets of different dimensions. It has also been compared to other mixup variants (manifold mixup) and on adversarial attacks. The results show a small increase in the performance. I think the empirical comparison is complete. However, I have some concerns regarding the theory as stated above. Indeed using minibatch OT as you do, creates non optimal connections between samples including connections between different examples [3]. Thus a longer discussion on minibatch OT is required.


Significance: The idea is interesting but is not too novel as the idea of coupling close samples has already been explored in the original mixup paper through k-nearest neighbourh. The difference is the use of optimal transport to determine how to connect samples instead of doing it randomly as done in the original mixup.

Related Work and Discussion: The strength of the methods are discuted but the aspect of minibatch OT is lacking and as such, the discussion is not complete in my opinion. The limitations of the methods is not discussed enough in my opinion. Authors could have empirically experimented the percentage of connections between different clusters of data to show that their method respects the manifold of data. Regarding related work, there are missing discussions which I think are important with previous methods (k nearest neighbourh from original mixup for instance).

Clarity: A reader not familiar with the mixup regularization could understand previous work as well as the presented method. The paper is easy to read and correctly written. The objective is clearly stated and I have not seen typos in the main text. I think that, from the text, one could reproduce the proposed methods. However, I also think that a discussion on how they create the full minibatch of data used during training should be included in the paper. Maybe adding examples or pseudo code is a good way to improve the clarity on these details.

Questions and remarks:

1. You do not really use optimal transport but minibatch optimal transport[1,2] and you should discuss the differences. As such, it is known from [3,4,5] that it creates non optimal connections between clusters of data due to the sampling of minibatches. Following your intuition to preserve the manifold structure, I wonder why all your best scores are not for the biggest k, as the biggest k would more preserve the structure. This is contradictory with the initial prediction. Finally, if k grows to infinity, the proposed method would just mixup samples between themselves, thus no doing any mixup at all... This is also a concerning point, a study focused on minibatch OT (what authors do in practice) might alleviate this problem.

2. Even if the connections between far away data are rare, it is noted in [5] that they happen and can harm the neural networks on some applications. I wonder their impact on the training. (see point 8 for a related point)

3. Furthermore, due to the possible imbalanced classes, the data structure you are looking for, might not be possible in practice even for big k and even with OT between the full distributions. How does your method work for highly imbalanced case ? May be using partial OT or unbalanced OT with your minibatch formulation might help.

4. I have some concerns with Section 3.1 and Section 3.2. You have applied your theory to exact OT, or minibatch OT with only one batch couple, while you are doing minibatch OT which is an expectation of optimal transport terms over minibatches of data. The latter favorises the creations of non optimal connections and is different of the former. You consider the case where the number of data grows to infinity but in practice you only use really small k. As such the theory is not in concordance with the practice and I am not convinced.

5. While the use of Optimal Transport is appeling and increases the scores, it is not the first time that doing miwup between similar data is used. Indeed, in the original mix up paper, authors tried to used a k-nearest neighbourh and did not see improvement over original mixup. A discussion between k-nearest neighbourh is thus lacking.

6. What is the full batch size you used for your training ? Was it k ?

7. Could you please share a pseudo algorithm of your method ? It could be in appendix and help to understand some details of your training.

8. As the motivation of your method is to respect the manifold of data, an interesting experiment, in my opinion, would be to measure the average percentage of connections between data which belong to different clusters.

[1] DeepJDOT: Deep Joint Distribution Optimal Transport for Unsupervised Domain Adaptation, Damadoran et al.

[2] Learning Generative Models with Sinkhorn Divergences, Genevay et al.

[3] Learning with minibatch Wasserstein: asymptotic and gradient properties, Fatras et al.

[4] Minibatch Optimal Transport distances; analysis and applications, Fatras et al.

[5] Unbalanced minibatch Optimal Transport; applications to Domain Adaptation, Fatras et al.


----------------------------------------------------------------------------------------------------------------------------------------------------------------
############################################ UPDATE ############################################
----------------------------------------------------------------------------------------------------------------------------------------------------------------
I have read your answers. While I think some elements have greatly improved, I still think that there are some problems. I think the discussion on the minibatch transport plan should be better discussed at least in supplementary. I have also a concern of the manifold structure preservation in high dimension. Most reviewers also think that the use of theorems in practice is unclear. For all these reasons I keep my score unchanged.

**Summary Of The Paper:**

Goals: The paper presents a new manner of mixing up training samples to create new training distributions like mix up. It uses optimal transport between minibatches of data of size k. Then they perform mixup between transported data and uses these new data in the training of their neural networks. The intuition is that their method is more able to respect the manifold of data contrary to original mixup which mix samples uniformly at random.


**Summary Of The Review:**

Recommendation: Reject. While I agree the idea is appealing, it is not too novel and some missing discussions would lead to a huge change in the original paper in my opinion.

---

> ### Author Response · Authors · 2021-11-16
> **Thank you, and responses**
>
> Thank you for the comments and close reading of the work. We especially appreciate the fact that you found our writing to be clear, and our empirical evaluation to be relatively complete. Here are some responses to your questions and remarks.
>
> 1. There may be a confusion here on the rationale for the use of OT in our regularization setup. We are using it to come up with perturbed vicinal datasets that better reflect the structure of the training dataset in a way that is quick, cheap, and easy to implement. We do this by taking batches of data of size $k$, randomly sampled, and coupling them through OT. The geometry of the training dataset is respected by the OT coupling, and the randomness introduces a non-deterministic nature to the process. In particular, we never require estimation of an OT distance. **In the applications mentioned by the reviewer**, the goal is to estimate the optimal transport distance between two different distributions, and a minibatch approach averages the result over many smaller OT problems between sampled batches from the source and target distributions. In our setting, the source and target distribution are the same: the training dataset, and the exact OT distance here is clearly 0. One could view our approach as minibatch optimal transport with batches of size $k$, but there is no averaging of transport costs or attempt to recover any sort of exact distance. We would be glad to note the connection to this terminology, and cite the sources mentioned. The training data is already randomized by the data loader, so to take a random subset of size $k$, we just select the next $k$ items that are presented.
>
> 2. It is true that it is not possible to guarantee avoidance of long-range connections due to the random sampling of batches, but they are by definition less likely with k-mixup than with 1-mixup. Additionally, given the large sizes of the datasets, law of large numbers begins to apply here and consistently adverse matches are virtually impossible. Finally,  a few random long-range matches can provide additional smoothness over large voids between data manifolds that may not otherwise be covered by the vicinal distribution and thus are not harmful when of low frequency.
>
> 3. Theorem 1 of our paper gives some characterization of the behavior expected in the unbalanced case in a particular clustered scenario. As the probability masses $p_i$ start to differ greatly, the number of cross-cluster identifications increase, but in the worst case, we still have that the fraction of cross-cluster identifications grows as $O(m/2k)$ with $m$ clusters. In essence, as each $k$-batch is sampled from the overall distribution, the proportions of classes/clusters in each $k$-batch should be similar to that of the training dataset overall. It is not clear to us that partial or unbalanced OT is necessary to deal with class imbalance, especially since both k-samples will have the same expected class imbalance.
>
> 4. The Theorems in 3.1 and 3.2 apply to a single optimal transport between batches, and try to show that the regularized vicinal datapoints will better reflect the overall structure of the data. As $k$-mixup is simply this process applied over the entire dataset partitioned into such batches, the theory applies to the vicinal dataset overall.
>
> 5. There is a clear difference between the use of optimal transport and k-nearest neighbors to generate vicinal distributions. This difference is clearly seen in the top right of Fig. 2, where we see that couplings are not always done to the nearest neighbors, but are done in a way that lowers the sum cost of the couplings together. This set of couplings better reflects manifold and cluster structure. We note also that use of a k-nearest neighbors strategy is unlikely to generate any cross-cluster interpolation, and misses out on smoothing label transitions in the intermediate areas. We would be happy to include more discussion of this in a revision.
>
> 6. Our batch sizes were 128 across all architectures and experiments, which is divisible by twice the “minibatch” sizes $k$ for all possible choices there. This ensures that minibatch couplings and vicinal datasets may be computed within single batches.
>
> 7. Minibatches are generated via the input stream. $k$ at a time are selected, which amounts to a random sample of subsets of size $k$. Pairs of such $k$-batches are paired with OT and a random interpolation parameter $\lambda$ is drawn from the beta distribution for generation of the vicinal distribution. We’ll include pseudocode in a revision.
>
> 8. Our theoretical results (Theorem 1, in particular) characterize this in a clustered setting. We’d be happy to run such experiments on toy models or smaller datasets for empirical validation.
>
> We hope these answers are convincing, and will be glad to clarify further if needed. If they bolster your estimation of the work, please consider modifying your score to reflect this.

---

> > ### Comment · Reviewer_P7xo · 2021-11-23
> > **Thank you for your detailed answers.**
> >
> > I have read your answers.
> >
> > 1. It is true that the references I gave applied minibatch OT on data fitting experiments *but* the authors of [3,4] also theoretically studied the minibatch transport plan, that you use in practice to match the source and target (which is a minibatch of the source) distributions. This is what I meant by my point and your case corresponds to the case where the number of batch couples equals to 1. I am not statisfied by the added discussion as I think the consequences on the transport plan should be discussed. Furthermore the sentence "Ours is the first application where the underlying source and target distribution are the same" is not correct as it was also done in previous work [6].
> >
> > 2. (and 8.) Thank you, this experiment is indeed very interesting in my opinion. I think it should be added to the manuscript to prove the motivation of your method (better respect manyfold structure). However, I am also even more convinced that a better discussion with [3,4] is now needed (see 1.) as a small batch size tends to create non optimal connections between clusters in the transport plan.
> >
> > 3. On a second thought, I agree that Unbalanced OT is not suited for your application. The purpose of using Partial OT would be to avoid to associate samples which are too far from each other for small k. But I agree it might not be necessary for large enough k.
> > 4. Ok I understand but in practice you always take a small minibatch size. I am thus not convinced by the theorems which might require a large k.
> > 5. Ok I agree with your point but please cite the original mixup paper in the new sentence as it was studied here.
> > 6-7. Thank you. The pseudo code is really helpful in my opinion.
> >
> > [6] Missing Data Imputation using Optimal Transport, Muzellec et al.

---

> > > ### Author Response · Authors · 2021-11-29
> > > **Thank you for reading, and further discussion**
> > >
> > > Thank you very much for reading our response and responding in kind. Some further questions:
> > >
> > > > It is true that the references I gave applied minibatch OT on data fitting experiments but the authors of [3,4] also theoretically studied the minibatch transport plan, that you use in practice to match the source and target (which is a minibatch of the source) distributions. This is what I meant by my point and your case corresponds to the case where the number of batch couples equals to 1. I am not statisfied by the added discussion as I think the consequences on the transport plan should be discussed.
> > >
> > > Can you elaborate on how you see these results being relevant to our scenario?
> > >
> > > Our understanding of [3,4] is that they are targeting estimation of OT distances to be used in objectives for training of GANs and domain adaptation. The network or model is aiming to map one distribution onto another where one would like to optimize the weights to achieve the best mapping possible, in the Wasserstein sense. As it's not computationally feasible to compute this objective over the entire source and target datasets, one samples "minibatches" and optimizes the weights with respect to this approximate objective (or averages over OT distance between several pairs of "minibatches").
> > >
> > > These papers prove facts about how well these minibatch distances approximate the true objective, how well the averaged transport plans approximate the true optimal transport, whether the minibatch distances provide unbiased gradients, and show that the minibatch distance is not a true distance.
> > >
> > > In our scenario, we draw two minibatches from the same distribution, the training set, and then couple them through optimal transport. In our method, we do not use the calculated OT distance between minibatches, nor do we take any sort of gradient with respect to this distance. We do use the resulting OT between minibatches to determine our perturbations, and results characterizing its closeness to the true transport (identity, in this case) could be relevant, but the papers only argue for bounds on the degree to which the averaged transport plans match the marginals, and these are not particularly useful for us.
> > >
> > > > Furthermore the sentence "Ours is the first application where the underlying source and target distribution are the same" is not correct as it was also done in previous work [6].
> > >
> > > The use of OT in this work is indeed closer in spirit to our use of it. The uses are distinct (data imputation vs. augmentation), but we'd be happy to include a reference to it. Thanks for noting it.
> > >
> > > > Ok I understand but in practice you always take a small minibatch size. I am thus not convinced by the theorems which might require a large k.
> > >
> > > These theorems are tackling a very challenging characterization to make in the general case, where there are a wide range of possible dataset geometries that might be encountered. This is the reason we appeal to large k: in order to make asymptotic statements about the general behavior expected with k-mixup. We feel that our empirical results with toy examples, Fig. 1 and 2, and with real datasets provide experimental evidence for preservation of cluster and manifold structure.
> > >
> > > > Ok I agree with your point but please cite the original mixup paper in the new sentence as it was studied here. 6-7. Thank you. The pseudo code is really helpful in my opinion.
> > >
> > > Glad to provide the pseudocode, and we can add in the citation.

---

> ### Author Response · Authors · 2021-11-19
> **A quick experiment response to point 8**
>
> For the toy datasets, we ran experiments measuring the average percentage of connections between data which belong to different clusters in minibatches of size 128 for $k = 1, 2, 4, 8, 16$.
>
> The means were computed from 235 sets of minibatches for each figure in the table.
>
> | $k$ | Swiss Roll | Four Bars | One Ring |
> | --- | --- | --- | --- |
> | 1 | 0.500 | 0.493 | 0.502 |
> | 2 | 0.489 | 0.467 | 0.457 |
> | 8 | 0.447 | 0.383 | 0.380 |
> | 4 | 0.475 | 0.429 | 0.291 |
> |16 | 0.397 | 0.311 | 0.205 |
>
> As can be seen the average number of cross-cluster connections decreases with increasing $k$, serving as evidence of better manifold structure preservation.

---

### Author Response · Authors · 2021-11-19
**Revision uploaded**

We have updated the paper to incorporate the revisions promised in our replies below to the reviewers, marking new material with "New" in the margin.

---

### Decision · Program_Chairs · 2022-01-20

**Decision:**

Reject

**Comment:**

This paper proposes an extension of mixup (a data augmentation method) to k-mixup using optimal transport. The idea is to select randomly at each iteration  two subsets of  k samples and compute the optimal transport solution. Each pairs of samples assigned by the optimal transport plan will then be used to perform mixup and promote smoothness in the prediction function. The authors also provide some theoretical results about preservation of the clusters. Finally numeric experiment show the interest of k-mixup on toy and real life dataset classification and study the effect of k and the $\alpha$ parameter (of the $\beta$ distribution).

All reviewers found the paper interesting and acknowledge that it leads to some performance improvements in practice. But they had several concerns that lead to low scores. The justification of the method an more specifically the link with the theoretical findings was found lacking, indeed the result make sens fr a large $k$ which is not was is done in practice (but experiments also show a decrease sometimes for large $k$). One interesting discussion  between the proposed approach and minibatch OT is also missing. In addition the reviewers found the numerical experiments interesting but regret that some mixup approaches have not been compared and also noted a small gap in performance for the proposed approach (with no variance reported). Also the Adversarial robustness measure is now considered weak in the literature and those results could have been made stronger with more modern adversaries. Their final concern was the fact that the method now has two parameters that needs tuning and that can have a large impact on the performance for limited gain. The authors did a detailed reply a,d edition of the paper that was very appreciated by the reviewers but that did not change their opinion that this paper still deserves some more work before being accepted.

For these reasons the AC recommend to reject the paper but strongly suggests that the authors take into account the reveiwers' comments before resubmitting to a ML venue.